# HRDE-2 drives small RNA specificity for the nuclear Argonaute protein HRDE-1

Shihui Chen [1] & Carolyn M. Phillips [1] ✉

RNA interference (RNAi) is a conserved gene silencing process that exists in diverse organisms to protect genome integrity and regulate gene expression. In *C. elegans*, the majority of RNAi pathway proteins localize to perinuclear, phase-separated germ granules, which are comprised of sub-domains referred to as P granules, *Mutator* foci, Z granules, and SIMR foci. However, the protein components and function of the newly discovered SIMR foci are unknown. Here we demonstrate that HRDE-2 localizes to SIMR foci and interacts with the germline nuclear Argonaute HRDE-1 in its small RNA unbound state. In the absence of HRDE-2, HRDE-1 exclusively loads CSR-class 22G-RNAs rather than WAGO-class 22G-RNAs, resulting in inappropriate H3K9me3 deposition on CSR-target genes. Thus, our study demonstrates that the recruitment of unloaded HRDE-1 to germ granules, mediated by HRDE-2, is critical to ensure that the correct small RNAs are used to guide nuclear RNA silencing in the *C. elegans* germline.

Animal genomes are frequently exposed to diverse damage, such as environmental pathogens, RNA viruses, and endogenous transposable elements (TE). One mechanism the nematode *Caenorhabditis elegans* employs to protect genome integrity is through the production of small interfering RNAs (siRNAs). When siRNAs are generated from an exogenous double-stranded RNA (dsRNA), cleavage of that dsRNA occurs through the activity of an endonuclease RNase III enzyme Dicer[1,2]. Small RNAs trigger gene silencing and finely tune gene expression; this silencing process is called RNA interference (RNAi)[3]. Small RNAs can also be generated endogenously from the genome, such as Piwi-interacting RNAs (piRNAs) and endogenous small interfering RNAs (endo-siRNAs).

Small RNAs do not work alone to silence genes, rather, they associate with their protein partner Argonaute (AGO) to form the RNA-induced silencing complex (RISC). In *C. elegans*, there are 27 Argonaute genes and 19 of them are known to encode functional Argonaute proteins[4,5]. Argonaute proteins contain conserved domains, including an N terminal domain, a PAZ domain, a MID domain, and a PIWI domain. The PAZ domain and MID domain contain small RNA binding pockets to stabilize the small RNA[6,7]. The PIWI domain contains an RNaseH domain that, in some cases, can cleave target RNAs. In *C. elegans*, many Argonaute proteins have lost the catalytic amino acids that make up the active site (DDH), and instead

they recruit downstream proteins to degrade mRNA by decapping and deadenylation, regulate translation, or modulate transcription through chromatin modification[4,5].

The *C. elegans* siRNA pathways can be broken down into two stages – primary siRNAs recognize deleterious RNAs and initiate amplification of secondary siRNAs, while the more abundant secondary siRNAs mediate gene regulation. There are multiple sources for primary siRNAs. Exogenous siRNAs, produced by Dicer cleavage of double-strand RNA, associate with the Argonaute RDE-1[8–10]. Primary endo-siRNAs, or 26G-RNAs because they are generally 26 nucleotides in length and start with a guanosine, are produced by Dicer in concert with the RNA-dependent RNA polymerase (RdRP) RRF-3 and associate with the Argonaute proteins ERGO-1 during oogenesis and ALG-3 and ALG-4 during spermatogenesis[11–17]. piRNAs, or 21U-RNAs due to their 21-nucleotide length and 5' uracil, are transcribed as individual RNA Polymerase II transcripts and processed at both 5' and 3' ends to their mature form and associate with the Piwi Argonaute PRG-1[18–26].

Following target gene recognition, primary small RNAs trigger recruitment of the RdRPs, EGO-1 and RRF-1, to generate additional secondary small RNAs required to modulate gene expression[20,21,27]. The secondary siRNAs, or 22G-RNAs, named for their 22-nucleotide length and 5' guanosine, require additional factors for their production such as the exonuclease MUT-7, the endonuclease RDE-8, the poly(UG)

[1]Department of Biological Sciences, University of Southern California, Los Angeles, CA 90089, USA. ✉e-mail: cphil@usc.edu

polymerase MUT-2/RDE-3, and the helicases MUT-14 and SMUT-1[18,28–34]. 22G-RNAs are then loaded onto secondary worm-specific Argonaute (WAGO) proteins to silence genes post-transcriptionally or co-transcriptionally. It is currently unclear how these different classes of small RNAs are precisely sorted and loaded to the correct Argonaute proteins.

Post-transcriptional silencing occurs in the cytoplasm, where Argonaute proteins cleave mature mRNAs directly or recruit downstream factors that degrade them or prevent their translation. Co-transcriptional silencing is mediated by nuclear Argonaute proteins and occurs in the nucleus, so it is also referred to as nuclear RNA interference (nuclear RNAi). HRDE-1/WAGO-9 and NRDE-3/WAGO-12 are the two nuclear RNAi Argonaute proteins that function in the germline and soma respectively[35,36]. They recruit the NRDE factors, including NRDE-1, NRDE-2, and NRDE-4, and the histone methyltransferase SET-25 and MET-2, to silence genes by depositing repressive trimethylation marks on Histone H3 at Lysine 9 (H3K9me3) and inhibiting RNA polymerase II elongation[35–42]. HRDE-1 shares targets largely with WAGO-1[42], indicating that the nuclear RNAi pathway is likely working in parallel with the cytoplasmic WAGO proteins to modulate expression of the same target genes. HRDE-1-mediated gene silencing can also be inherited across multiple generations, leading to a heritable memory of both endogenous and exogenous RNA silencing[35]. When *hrde-1* mutants are raised at elevated temperature (25 °C), this failure in inheritance of small RNAs across generations results in a disruption of germ cell function triggering sterility of the animals within 3-5 generations[35]. In addition to silencing genes, small RNAs can also license gene expression. A special Argonaute protein CSR-1 protects genes from silencing by WAGO proteins. CSR-1 binds 22G-RNAs produced by the RdRP EGO-1 that targets germline protein-coding genes[31,43]. The precise details of CSR-1 22G-RNA biogenesis and gene licensing mechanisms are still unknown.

In the *C. elegans* germline, many of the small RNA pathway components reside in the cytoplasm and at perinuclear germ granules. Germ granules, or nuage, are phase-separated membraneless condensates that comprise proteins and RNAs. Many proteins associated with germ granules possess intrinsically disordered regions (IDRs), which mediate multivalent protein-protein interaction and protein-RNA interaction to drive granule assembly[44–46]. The *C. elegans* germ granules can be divided into multiple subdomains possessing distinct protein components and presumably unique molecular functions. Currently, we know of four distinct domains, P granules, Z granules, *Mutator* foci, and SIMR foci[33,47–50], but there may be more still to discover. P granules are thought to be sites of mRNA surveillance and tightly associated with nuclear pores to survey transcripts as they exit the nucleus[51]; Z granules are known to mediate transgenerational inheritance[48,52]; *Mutator* foci contain factors required for 22G-RNA amplification[33]; and SIMR foci, the most recently discovered subdomain, mediate 22G-RNA production at piRNA target genes[50]. SIMR foci have only one well-characterized component, SIMR-1, which is a Tudor domain protein[50]. Thus, the full complement of protein components and detailed molecular processes occurring in SIMR foci remain a mystery.

An important open question in the *C. elegans* small RNA field centers around the means by which various classes of 22G-RNAs are sorted into the correct Argonaute proteins. In other systems, numerous mechanisms have been identified that promote small RNA sorting, but they vary from Argonaute to Argonaute and from organism to organism. For example, *Arabidopsis* Argonaute proteins have a strong preference for specific 5′ nucleotides, to the extent that changing the 5′ nucleotide can redirect a small RNA into a different Argonaute complex[53–55]. But with 10 *Arabidopsis* Argonaute proteins and only four nucleotides, 5′ nucleotide preference cannot be the sole driver of specificity. Alternate and hierarchical mechanisms by which Argonaute proteins derive specificity include protein-protein interactions with

biogenesis and loading factors, structure and thermodynamic properties of their duplexed precursors, small RNA length, and differential stability of the small RNA post-Argonaute binding, which is dependent on complementarity of target binding and protection from degradation mediated by small RNA 2′-O-methylation[56–63]. In *C. elegans*, Argonaute specificity between WAGO-class and CSR-class 22G-RNAs cannot be dictated by properties of duplexed precursors or degree of 2′-O-methylation, since these classes of small RNAs are synthesized as single-strand RNAs by RdRPs and neither is methylated[64,65]. Specificity also cannot be dictated by 5′ nucleotide, length, unique tissue-specific or developmental expression, or degree of complementarity to target RNAs, because these properties are identical between the CSR- and WAGO-classes of siRNAs[31,43]. Further, no loading factors have been identified for either CSR-1 or the WAGO clade of Argonaute proteins and the RdRP EGO-1 can synthesize both types of 22G-RNAs.

Here, we have used multiple methods with the initial aim of discovering new SIMR foci components, but that ultimately led to the discovery of a mechanism by which WAGO-class and CSR-class 22G-RNAs are correctly sorted into their respective Argonaute proteins. First, we determined that HRDE-2 interacts with SIMR-1 by yeast two-hybrid (Y2H) screening. HRDE-2 colocalizes with SIMR-1 at SIMR foci and its germ granule association is independent of *Mutator* foci. Next, using immunoprecipitation-mass spectrometry, we found that HRDE-2 interacts with HRDE-1, the germline-expressed nuclear Argonaute protein. While this result was initially surprising because HRDE-1 is known to be nuclear[35], localization analysis demonstrated that HRDE-1 localizes to germ granules only when unbound to 22G-RNAs. Further, HRDE-2 is required for the germ granule association of unbound HRDE-1 and is essential to promote loading of the preferred WAGO-class 22G-RNAs into HRDE-1. In the absence of HRDE-2, HRDE-1 is incorrectly loaded with CSR-class 22G-RNAs that are sufficient to promote transcriptional silencing at some germline expressed genes. Therefore, in this study we have identified a protein, HRDE-2, that is critical for promoting small RNA specificity and germ granule association of the Argonaute HRDE-1, ultimately affecting selection of HRDE-1 target genes and deposition of H3K9me3 at these ectopic HRDE-1 target-gene loci.

## Results

### HRDE-2 interacts with SIMR-1 and associates with SIMR foci

To date, no protein interactors of SIMR-1 have been identified. To identify proteins that interact with SIMR-1, we conducted a yeast two-hybrid (Y2H) screen with SIMR-1 and detected multiple potential interactions (Supplementary Data 1). The top three interactors captured were HRDE-2, an uncharacterized protein F45D3.4, and the kinase MBK-2 (Fig. 1a). The HRDE-2 protein was captured at the highest frequency in the yeast two-hybrid screen and was previously known to have a role in the small RNA pathway[66,67], so we chose to initially focus on the potential interaction between HRDE-2 and SIMR-1. HRDE-2 was first identified in a genetic screen for RNAi inheritance factors; it was shown to function in the same genetic pathway as the nuclear Argonaute HRDE-1 and is required for HRDE-1 to bind siRNAs following exogenous RNAi[66].

To identify domains present in the HRDE-2 protein sequence, we used the HHpred server to identify remote homologs with similar 3D structure[68,69]. We found that the N-terminal portion of HRDE-2 contains structural similarity to the C-terminal domain of helicases of the SF1 and SF2 superfamilies (HELICc) (Fig. 1b). We then used PONDR[70] to predict disordered regions within the protein, and found that the C-terminal region of HRDE-2 is disordered (Fig. 1b).

Previous work demonstrated that HRDE-2 localizes to germ granules[66,67], but the precise germ granule compartment to which HRDE-2 localized was unknown. First, to determine whether the N-terminal HELICc domain is important for HRDE-2 function, we deleted this region (a.a. 26-178) and observed that HRDE-2 no longer

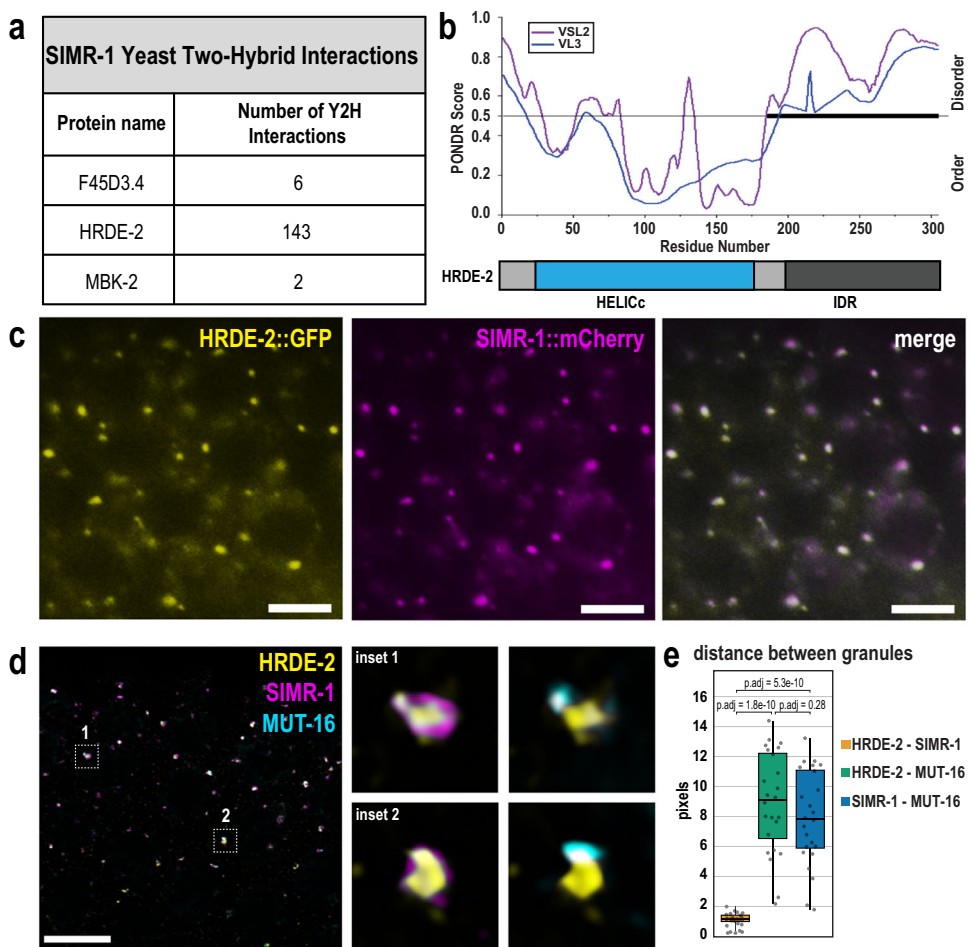

**Fig. 1 | The IDR and HELICc domain-containing protein, HRDE-2, localizes to SIMR foci. a** Medium and high-confidence protein interactions identified by a SIMR-1 yeast two-hybrid screen. See Supplementary Data 1 for a complete list of interacting proteins. **b** Graph displaying disorder tendency for the HRDE-2 protein sequence. The disorder prediction was made using PONDR (http://www.pondr.com) with VSL2 and VSL3 parameters. Regions with PONDR scores of greater than 0.5 indicate disorder and regions with PONDR score less than 0.5 indicate order. The bold line from a.a. 185 to 305 indicates the longest predicted disordered region of HRDE-2. Below, a domain diagram for HRDE-2 shows that it contains an N-terminal HELICc domain and a C-terminal intrinsically disordered region. **c** Live imaging of SIMR-1::mCherry::2xHA (magenta); HRDE-2::2xTy1::GFP (yellow) hermaphrodites, showing that SIMR-1 and HRDE-2 colocalize. At least ten individual germlines were imaged. Scale bars, 5 μM. **d** Confocal imaging of a SIMR-1::mCherry::2xHA MUT-16::3xMyc; HRDE-2::2xTy1::GFP in a dissected day-one adult germline, using antibodies against GFP, HA, and Myc. Insets, at right, show visual examples of SIMR-1 proximity to HRDE-2 compared to MUT-16 proximity to HRDE-2. All images are a single focal plane. At least five individual germlines were imaged. Scale bar, 5 μM. **e** Box plot showing distance between the centers of fluorescence for HRDE-2, SIMR-1, and MUT-16 granules. 24 individual granules from 4 germ cells of 2 animals were used for quantification. Measurements for each of the 24 granules are shown as individual dots. Bolded midline indicates median value, box indicates the first and third quartiles, and whiskers represent the most extreme data points within 1.5 times the interquartile range, excluding outliers. Two-tailed *t*-tests were performed to determine statistical significance and *p* values were adjusted for multiple comparisons. See Methods for detailed description of quantification methods. Source data are provided as a Source Data file.

associates with germ granules (Supplementary Fig. 1a), indicating that the HELICc region is important for HRDE-2 to be assembled in germ granules. Next, to identify the subdomain of the germ granule to which HRDE-2 localizes, we examined whether HRDE-2 co-localizes with proteins found in each of the germ granule subdomains. Because HRDE-2 was initially identified through the yeast two-hybrid screen of SIMR-1, we first examined the localization of HRDE-2 relative to SIMR-1 by live imaging of the pachytene region in adult hermaphrodites and found the localization of the two proteins to be nearly identical (Fig. 1c). To further confirm that they colocalize, we performed immunofluorescence microscopy for SIMR-1, HRDE-2, and MUT-16, as MUT-16 marks *Mutator* foci which constitute a closely associated but separate germ granule subdomain from SIMR foci[33,50]. As previously demonstrated, we observed that SIMR-1 and MUT-16 localize near one another but are not perfectly co-localized (Fig. 1d, e). Similar results were observed for HRDE-2 and MUT-16 (Fig. 1d, e). In contrast, we found that HRDE-2 and SIMR-1 were highly coincident (Fig. 1d, e). We

also examined HRDE-2 localization in a *mut-16* mutant, where the *Mutator* foci fail to assemble in the absence of the scaffolding protein MUT-16[33]. We observed that HRDE-2 still localizes to perinuclear germ granules (Supplementary Fig. 1b), indicating that HRDE-2 localizes independently of *Mutator* foci, similar to what was previously observed for SIMR-1[50]. Lastly, to compare HRDE-2 localization to that of Z granules and P granules, we performed immunofluorescence microscopy for HRDE-2, ZNFX-1, and PGL-1, and found that HRDE-2 does not colocalize with either ZNFX-1 or PGL-1, indicating that HRDE-2 does not associate with either the Z or P granule compartments. (Supplementary Fig. 1c). Together, these results indicate that HRDE-2 is likely a component of the SIMR foci.

We next sought to address whether HRDE-2 and SIMR-1 depend on one another for localization. To this end, we introduced fluorescently-tagged SIMR-1 into a *hrde-2* mutant and fluorescently-tagged HRDE-2 into a *simr-1* mutant. We found that HRDE-2 still localizes to germ granules in the absence of SIMR-1, but the fluorescence intensity is

modestly reduced (Supplementary Fig. 1d). SIMR-1 also localizes to germ granules with wild-type expression levels in the absence of HRDE-2 (Supplementary Fig. 1e). These data indicate that SIMR-1 and HRDE-2 localize to germ granules independently of one another and neither is clearly acting as a scaffolding protein for SIMR foci assembly. We next performed western blot analysis on these mutants but we did not observe prominent protein expression changes for HRDE-2 in the *simr-1* mutant or the *mut-16* mutant; SIMR-1 expression level also remains the same in the absence of HRDE-2 (Supplementary Fig. 1f). These data indicate that neither HRDE-2 nor SIMR-1 are required for one another's expression or localization to germ granules.

### HRDE-2 interacts with nuclear RNAi protein HRDE-1

To further investigate the interaction between HRDE-2 and SIMR-1, we conducted a co-immunoprecipitation (co-IP) between SIMR-1 and HRDE-2. Surprisingly, we were unable to detect the interaction following immunoprecipitation of SIMR-1 and probing for HRDE-2, or vice versa, with our immunoprecipitation conditions (Supplementary Fig. 2a, b). At this point, it remains possible that the interaction between HRDE-2 and SIMR-1 is either transient or unstable. Alternatively, it is also possible that they do not interact directly in vivo.

Next, we sought to identify additional interactors of either HRDE-2 or SIMR-1. We performed IP mass-spectrometry (mass-spec) on SIMR-1, HRDE-2, and wild-type controls. Consistent with the co-IP, SIMR-1 was not captured in the HRDE-2 mass-spec experiment and HRDE-2 was not captured in the SIMR-1 mass-spec experiment (Fig. 2a). Interestingly, the most prominent HRDE-2 interacting protein was the nuclear Argonaute protein, HRDE-1 (3/3 replicates), while another nuclear Argonaute protein, NRDE-3, was identified as an interactor of SIMR-1 (2/3 replicates) (Fig. 2a, Supplementary Fig. 2c, Supplementary Data 1). The interaction between HRDE-2 and HRDE-1 was initially surprising, because HRDE-2 localizes to germ granules in the cytoplasm while HRDE-1 localizes to the nucleus. However, we were able to further confirm this interaction by co-IP (Fig. 2b). In a recent publication, IP-mass spec of NRDE-3 from embryos lysates revealed the reciprocal interaction between NRDE-3 and SIMR-1[67], confirming their association as interacting proteins. Further characterization of the SIMR-1 and NRDE-3 interaction will be reported elsewhere. Therefore, our IP mass-spectrometry and co-IP results support that HRDE-2 interacts with the germline-expressed nuclear Argonaute HRDE-1 while SIMR-1 interacts with somatic nuclear Argonaute NRDE-3.

### Unloaded HRDE-1 translocates to germ granules

Given that HRDE-2 interacts with the nuclear Argonaute protein HRDE-1, we next investigated whether loss of HRDE-2 affects the nuclear localization of HRDE-1. Wild-type HRDE-1 localizes to the nucleus of germ cells in one-day-old adults, as expected (Fig. 3a)[35]. However, in addition to the nuclear signal, we also observed a small amount of germ granule localization exclusively in the diplotene region of the germline of L4 stage animals (Supplementary Fig. 3a). When we introduced a *hrde-2* mutation, we observed no change in HRDE-1 localization, with HRDE-1 still localized to the nucleus (Fig. 3a). HRDE-1 localization in a *simr-1* mutant shows similar nuclear localization (Supplementary Fig. 3a). Because the HRDE-2 paralogs ENRI-1 and ENRI-2 are thought to bind to unloaded NRDE-3 and prevent premature entry of NRDE-3 to the nucleus[67], we wanted to examine HRDE-1 localization in a situation where HRDE-1 is not localized exclusively to the nucleus. We hypothesized that, similar to NRDE-3[36], HRDE-1 nuclear localization might be regulated by small RNA binding. To this end, we introduced a *mut-2/rde-3* mutant, which depletes WAGO class 22G-RNAs[71,72], into the tagged HRDE-1 strain. We observed that in the *mut-2* mutant, HRDE-1 shows prominent perinuclear germ granule localization, in addition to its nuclear localization (Fig. 3a). This germ granule association in the *mut-2* mutant requires HRDE-2, as HRDE-1 expression is entirely nuclear in the *mut-2; hrde-2* double mutant (Fig. 3a). These

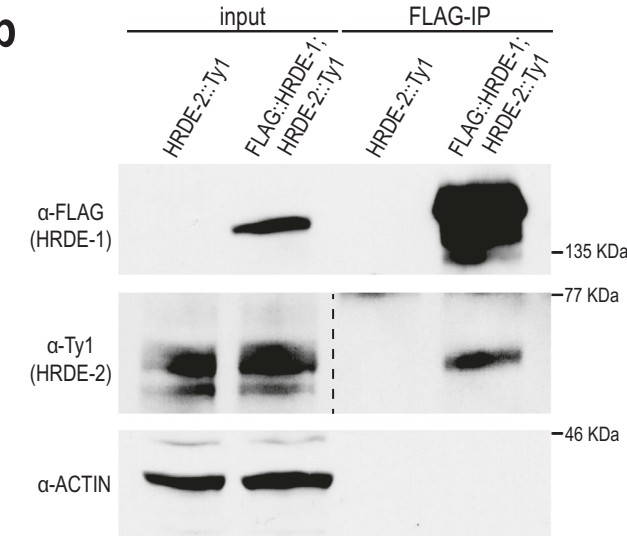

**a** Peptide counts from IP-mass spec

| Protein Name | wild-type IP | SIMR-1 IP | HRDE-2 IP |
|---|---|---|---|
| SIMR-1 | 0 | 69 | 0 |
| HRDE-2 | 0 | 0 | 35 |
| HRDE-1 | 0 | 0 | 38 |
| NRDE-3 | 0 | 31 | 0 |

**Fig. 2 | The germline-expressed nuclear Argonaute HRDE-1 interacts with HRDE-2. a** Peptide counts for proteins identified by the IP-mass spec of wild-type, SIMR-1::mCherry::2xHA, and HRDE-2::2xTy1::GFP strains, replicate 1. Peptide counts for two additional replicates can be found in Supplementary Fig. 2c. Anti-HA conjugated beads were used for SIMR-1 IP and wild-type IP, and anti-GFP conjugated beads were used for HRDE-2 IP and wild-type IP. The IPs with anti-HA and anti-GFP beads from wild-type samples had the same results (0 peptide count for the SIMR-1, HRDE-2, HRDE-1, and NRDE-3 proteins). Full protein lists are in Supplementary Data 1. **b** Western blot of HRDE-2 and HRDE-1 following HRDE-1 immunoprecipitation from the GFP::3xFLAG::HRDE-1; HRDE-2::2xTy1::GFP strain, shows that HRDE-2 interacts with HRDE-1. The HRDE-2::2xTy1::GFP strain was used as negative control. Anti-FLAG conjugated beads were used to immunoprecipitate GFP::3XFLAG::HRDE-1. Anti-FLAG, anti-Ty1, and anti-Actin antibodies were used to detect HRDE-1, HRDE-2, and Actin, respectively. This co-immunoprecipitation and western blot have been repeated three times with the same result. Source data are provided as a Source Data file.

data indicate that HRDE-2 can recruit HRDE-1 to germ granules in the absence of its preferred small RNA partners. Further, in the absence of HRDE-2, HRDE-1 becomes fully nuclear again regardless as to whether its preferred small RNAs are present.

We additionally compared the HRDE-1 localization in a *mut-2* mutant, to the *simr-1 mut-2* double mutant. We observed that in the absence of both SIMR-1 and MUT-2, HRDE-1 is primarily localized to the nucleus, though there is some weak granule localization (Supplementary Fig. 3a). These data indicate that SIMR-1 is also required for the germ granule retention of HRDE-1 in a *mut-2* mutant, but we cannot separate whether this is a direct effect or whether it is due to the modestly reduced germ granule localization of HRDE-2 in the *simr-1* mutant (Supplementary Fig. 1d).

We have two hypotheses to explain why HRDE-1 returns fully to the nucleus in a *mut-2; hrde-2* double mutant – 1) HRDE-2 prevents unloaded HRDE-1 from entering the nucleus, or 2) HRDE-2 promotes loading of HRDE-1 with the correct small RNAs. In the first hypothesis, the HRDE-1 found in germ granules in the *mut-2* mutant is unloaded with small RNAs and, in the *mut-2; hrde-2* double mutant, the unloaded HRDE-1 is able to enter the nucleus. In this scenario, HRDE-2 acts to

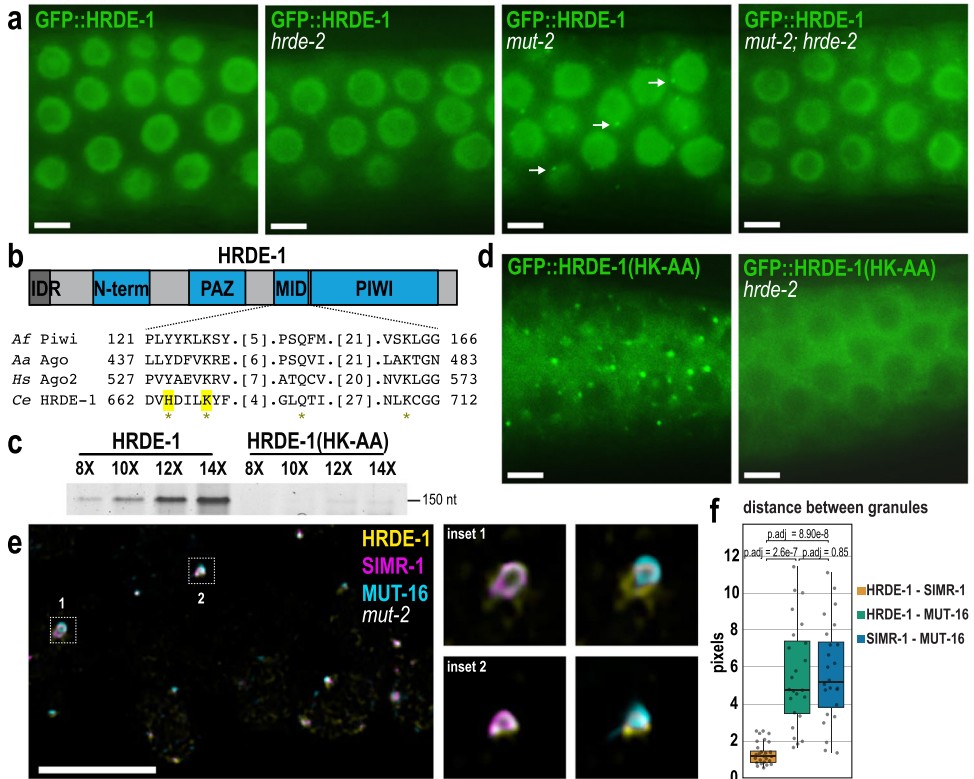

**Fig. 3 | Unloaded HRDE-1 localizes to SIMR foci. a** Live imaging of GFP::3xFLAG::HRDE-1 in wild-type, *hrde-2*, *mut-2*, and *mut-2; hrde-2* mutants. Arrows indicate perinuclear HRDE-1 granules visible only in *mut-2* single mutants. **b** Diagram of HRDE-1 protein domains. HRDE-1 includes an N-terminal intrinsically disordered region (IDR), PAZ, MID, and PIWI domains. Below, sequence alignment of the conserved 5′-phosphate binding residues across various species of Argonaute proteins. Prefix *Af, Archaeoglobus fulgidus; Aa, Aquifex aeolicus; Hs, human; Ce, Caenorhabditis elegans*. Asterisk indicates conserved Y-K-Q-K. The highlighted amino acids in HRDE-1 are mutated to alanine to generate the HRDE-1(HK-AA) small RNA-binding mutant. **c** Small RNA binding assay in HRDE-1 and HRDE-1(HK-AA) small RNA-binding mutant. The assay was performed using different PCR cycles (8X, 10X, 12X, 14X) to assess the amount of small RNA recovered from a HRDE-1 IP. This small RNA binding assay has been repeated three times with the same result. **d** Live imaging of GFP::3xFLAG::HRDE-1(HK-AA) in wild-type and *hrde-2* mutant strains. Small RNA-binding defective HRDE-1 is excluded from the nucleus and recruited to germ granules, and HRDE-2 is required for germ granule association.

**e** Confocal imaging of SIMR-1::mCherry::2xHA, MUT-16::3xMyc, and GFP::3xFLAG::HRDE-1 in a dissected germline of a day-one adult *mut-2* mutant, using antibodies against FLAG, HA, and MYC. Insets, at right, show examples of SIMR-1 and MUT-16 proximity to HRDE-1. Images are a single focal plane. **f** Box plot showing distance between the centers of fluorescence for HRDE-1, SIMR-1, and MUT-16 granules. Measurements for each of the 24 granules are shown as individual dots. Bolded midline indicates median value, box indicates the first and third quartiles, and whiskers represent the most extreme data points within 1.5 times the interquartile range, excluding outliers. Two-tailed *t*-tests were performed to determine statistical significance and *p* values were adjusted for multiple comparisons. All microscopy images are from the pachytene region of day-one adult animals and individually adjusted for brightness and contrast to optimize visualization of germ granules or lack thereof. At least five individual germlines were imaged for each strain. All scale bars, 5 µM. Source data are provided as a Source Data file.

prevent unloaded HRDE-1 from entering the nucleus. Alternatively, in the second hypothesis, the HRDE-1 found in germ granules in the *mut-2* mutant is still unloaded with small RNAs, but in the *mut-2; hrde-2* double mutant, HRDE-1 is now mis-loaded with incorrect small RNAs, which allowed it to translocate to the nucleus, with nuclear import still requiring small RNA binding. In this scenario, HRDE-2 acts to promote correct small RNA loading for HRDE-1 but plays no direct role in regulating the nuclear import of HRDE-1.

To address these possibilities, we generated a small RNA binding-defective HRDE-1 strain, where conserved residues in the MID domain that bind the 5′ end of the small RNA have been mutated to abolish its small RNA binding capability (H664A and K668A, henceforth referred to as HRDE-1(HK-AA)) (Fig. 3b)[6]. To confirm that the HRDE-1(HK-AA) strain is in fact small RNA-binding defective, we performed a small RNA binding assay on wild-type HRDE-1 and HRDE-1(HK-AA) strains. We found that wild-type HRDE-1-bound small RNAs are approximately 20-fold more abundant than HRDE-1(HK-AA)-bound small RNAs (or ~2.5-fold when normalized to HRDE-1 protein levels, which are reduced in the HRDE-1(HK-AA) mutant (Fig. 4c)), supporting that the HRDE-1(HK-AA) strain is small RNA binding-defective (Fig. 3c).

First, to establish whether HRDE-1 is recruited to germ granules when unloaded, we examined the localization of HRDE-1(HK-AA) and observed abundant perinuclear granules (Fig. 3d). However, it is important to note that, unlike the *mut-2* mutant where HRDE-1 was localized both to the nucleus and the germ granules, the HRDE-1(HK-AA) mutant was exclusively cytoplasmic and germ granule-associated. These data support the idea that unloaded HRDE-1 does not enter the nucleus. In the *hrde-2* mutant, HRDE-1(HK-AA) remains cytoplasmic but becomes dispersed from the granules (Fig. 3d), demonstrating that HRDE-2 is required to retain unloaded HRDE-1 in germ granules. These data also indicate that small RNA binding-defective HRDE-1 (HRDE-1(HK-AA)) cannot enter the nucleus even in the absence of HRDE-2, disproving the first hypothesis described above. Thus, our data support the model that HRDE-2 may promote correct loading of small RNAs into HRDE-1.

**Unloaded HRDE-1 associated with SIMR foci**

To interrogate which germ granule subdomain unloaded HRDE-1 associates with, we performed immunofluorescence microscopy on HRDE-1 in a *mut-2* mutant background in combination with tagged

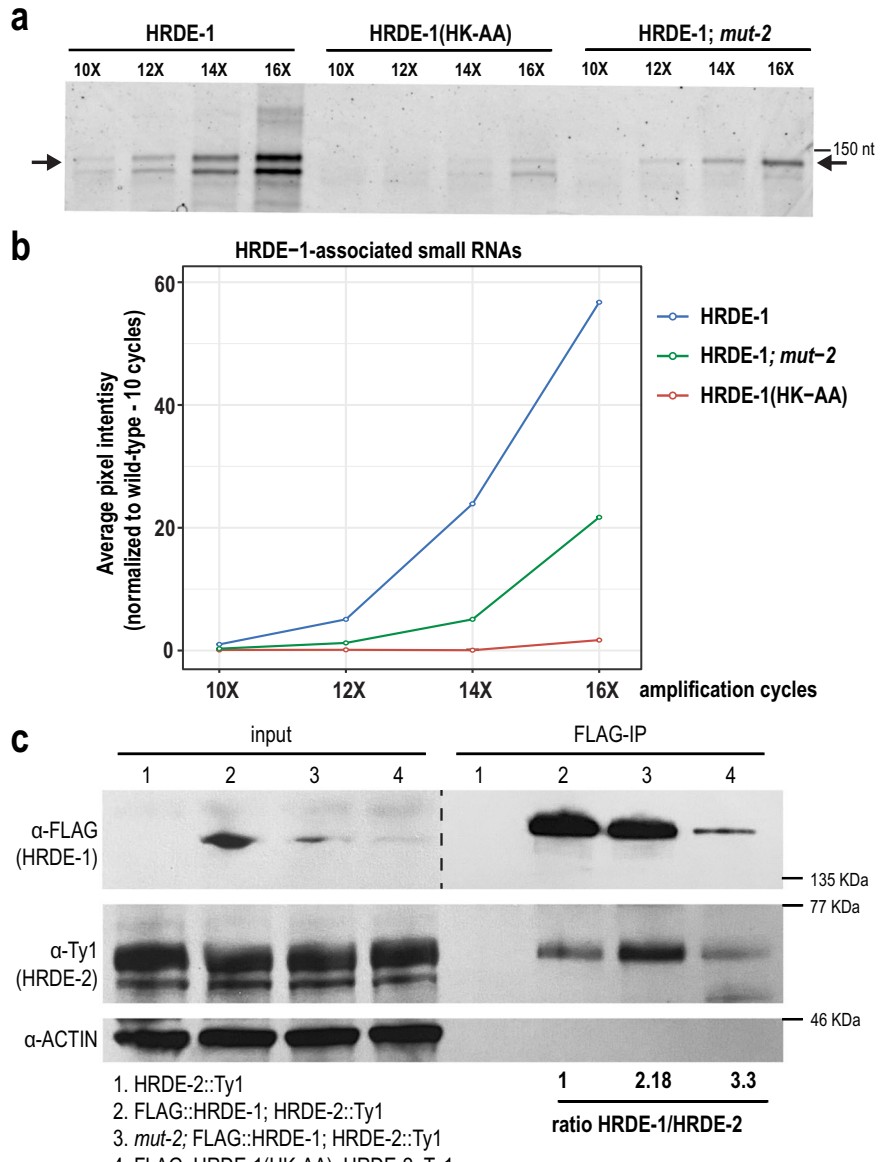

**Fig. 4 | HRDE-1-HRDE-2 interaction is modulated by small RNA binding. a** Small RNA binding assay following immunoprecipitation of HRDE-1, HRDE-1(HK-AA), and HRDE-1 in a *mut-2* mutant. The assay utilized different PCR cycles (10X, 12X, 14X, 16X) to assess the amount of small RNAs immunoprecipitating with HRDE-1. Arrows point to the band of the correct size for small RNA plus ligated adapters. Detailed protocol for the small RNA binding assay can be found in the Methods. **b** Quantification of HRDE-1-bound small RNAs from the small RNA binding assay. Average pixel intensities from two biological replicates are normalized to wild-type 10X PCR cycles. **c.** Western blot following immunoprecipitation of HRDE-1 in wild-type, *mut-2* mutants, and *hrde-1(HK-AA)* mutants, where HRDE-1 was tagged with GFP::3xFLAG and HRDE-2 was tagged with 2xTy1::GFP. The HRDE-2::2xTy1::GFP strain was used as a negative control. Anti-FLAG conjugated beads were used to immunoprecipitate GFP::3xFLAG::HRDE-1 and anti-FLAG, Anti-Ty1, and anti-Actin antibodies were used to detect HRDE-1, HRDE-2, and Actin. Band intensities were quantified using ImageJ and normalized to wild-type (lane 2) to determine how the HRDE-1-HRDE-2 interaction is affected by the *mut-2* and *hrde-1(HK-AA)* mutants. In this replicate, *mut-2* and *hrde-1(HK-AA)* mutants lead to 2.18-fold and 3.3-fold increase in HRDE-1-HRDE-2 interaction, respectively. This co-immunoprecipitation and western blot have been repeated three times with similar results. Additional replicates are shown in Supplementary Fig. 4. Source data are provided as a Source Data file.

proteins from each of the other germ granule compartments. Our findings indicate that HRDE-1 foci are positioned adjacent to P granules and Z granules, but do not colocalize with them (Supplementary Fig. 3b). Both *Mutator* foci and SIMR foci were closely associated with HRDE-1 foci. After quantifying the 3D granule distance using ImageJ, we determined that HRDE-1 is more closely associated with SIMR-1 compared to MUT-16 (Fig. 3e, f). To confirm this result, we abolished *Mutator* foci by introducing a *mut-16* mutation into the HRDE-1(HK-AA) strain and found that small RNA binding-defective HRDE-1 still localizes to germ granules in the absence of *Mutator* foci (Supplementary Fig. 3c). We do not know the identity of any protein that is essential for SIMR foci integrity, however, we did observe that in the *simr-1* mutant, HRDE-1(HK-AA) granule intensity is reduced (Supplementary Fig. 3c). Together, these data suggest that granule-localized HRDE-1 associates with SIMR foci.

We next sought to identify the domains that are required for the granule and nuclear localization of HRDE-1. To this end, we used PONDR[70] to predict disordered regions within the HRDE-1 protein and cNLS Mapper[73], along with alignment to the NRDE-3 protein sequence, to identify the HRDE-1 nuclear localization signal (NLS) (Supplementary Fig. 3d). We identified a 63 amino acid IDR at the N-terminus of HRDE-1 and within that, an NLS at amino acids 34-36 (KRS) which aligns

with the previously identified NRDE-3 NLS (Supplementary Fig. 3d)[36]. We next generated the HRDE-1(KRS-AAA) NLS mutant and an IDR deletion mutant. By live imaging both strains, we found that HRDE-1 localizes primarily to the cytoplasm, with some weak germ granule localization, in the absence of either the NLS or the full IDR (Supplementary Fig. 3e). By western blot, the protein expression level is similar to wild-type HRDE-1 (Supplementary Fig. 3f). The cytoplasmic localization of HRDE-1(KRS-AAA) NLS mutant is consistent with the observation that an NLS mutant of NRDE-3 similarly localizes to the cytoplasm of somatic cells[36]. Moreover, the NRDE-3 NLS mutant still binds small RNAs, indicating that small RNA loading is independent of nuclear localization. To further address whether the HRDE-1(KRS-AAA) NLS mutant is loaded with small RNAs, we disrupted its ability to bind small RNAs by introducing the HK-AA small RNA binding mutant. As we predicted, the HRDE-1(KRS-AAA; HK-AA) mutant, lacking both the ability to bind small RNAs and the NLS, associated with germ granules (Supplementary Fig. 3e), similar to the HRDE-1(HK-AA) small RNA binding mutant (Fig. 3d). Together, these data indicate that the HRDE-1(KRS-AAA) NLS mutant is able to interact with HRDE-2 in germ granules when unloaded, and when loaded disperses to the cytoplasm, rather than the nucleus, due to loss of the NLS signal.

## HRDE-2 interacts more strongly with unloaded HRDE-1

To further investigate whether the differences in localization between HRDE-1 in the absence of WAGO-class 22G-RNAs (mut-2 mutant) and small RNA binding-defective HRDE-1 (HRDE-1(HK-AA)) are due to loading of incorrect small RNAs into HRDE-1 in the mut-2 mutant, we sought to assess the level of small RNAs loaded by HRDE-1 in each strain compared to wild-type. We found that the level of small RNAs loaded by HRDE-1 in the mut-2 mutant was reduced ~4 fold, compared to wild-type (Fig. 4a, b). Given that WAGO-class 22G-RNAs are reduced by approximately 90% in mut-2 mutants[72], these data further support our hypothesis that HRDE-1 may have the capacity to bind non-WAGO-class RNAs when WAGO-class 22G-RNAs are depleted.

We next sought to determine whether the interaction between HRDE-1 and HRDE-2 is occurring primarily in germ granules, where HRDE-1 is unloaded. We hypothesized that mutants with a greater proportion of HRDE-1 in germ granules (i.e. HRDE-1(HK-AA) or HRDE-1 in a mut-2 mutant) would interact more strongly with HRDE-2 than HRDE-1 in a wild-type background. To this end, we performed a co-IP between HRDE-1 and HRDE-2 in wild-type and each of these mutant backgrounds. We observed that, when normalizing for the amount of HRDE-1 immunoprecipitated from each sample, HRDE-2 interacts more strongly with HRDE-1 in both HRDE-1(HK-AA) and HRDE-1 in the mut-2 mutant compared to wild-type (~2.18 fold stronger than wild-type in the mut-2 mutant and ~3.3 fold stronger than wild-type in HRDE-1(HK-AA)) (Fig. 4c). Two additional replicates similarly showed stronger enrichment for HRDE-1(HK-AA) compared to wild-type (Supplementary Fig. 4a, b). There was more variability for the mut-2 mutant between replicates, which may be because HRDE-1 is moving between the nucleus and granule in the mut-2 mutant. We additionally performed IP mass-spec for HRDE-1 in wild-type, mut-2 mutant, and the hrde-1(HK-AA) mutant and found that more HRDE-2 peptides were found in the HRDE-1 immunoprecipitation from HRDE-1(HK-AA) compared to wild-type HRDE-1 (Supplementary Fig. 4c, Supplementary Data 1). We conclude that the interaction between HRDE-1 and HRDE-2 likely occurs when HRDE-1 is unloaded and localized to germ granules.

## HRDE-2 contributes to HRDE-1 small RNA preference

To directly address whether HRDE-2 is required for correct loading of small RNAs into HRDE-1, we immunoprecipitated HRDE-1 followed by high throughput sequencing of associated small RNAs in wild-type, hrde-2, mut-2, simr-1, mut-2; hrde-2 and simr-1 mut-2 mutant backgrounds. We performed worm growth, immunoprecipitation, and small RNA library construction in triplicate for each strain. By examining the small RNA distribution for each library, we confirmed that replicates show similar levels of small RNAs mapping to CSR- and WAGO-target genes but that these levels changed significantly between mutant and wild-type strains in input and IP samples (Supplementary Fig. 5a).

By comparing the levels of small RNAs mapping to each gene in the input and IP samples, we found that HRDE-1 in a wild-type background preferentially associates with small RNAs mapping to genes previously defined as WAGO-target genes, as expected (Fig. 5a, b and Supplementary Fig. 5a, b). When we immunoprecipitated HRDE-1 from hrde-2 mutant animals, we observed a striking change in the HRDE-1-bound small RNAs, where HRDE-1 now preferentially associates with small RNAs mapping to genes previously defined as CSR targets[74]. Even in a mut-2 mutant, where WAGO-class small RNAs are depleted by 86.2%, HRDE-1 is strongly associated with the remaining small RNAs mapping to WAGO-target genes, and this enrichment is lost in the mut-2; hrde-2 double mutant (Fig. 5a, b and Supplementary Fig. 5a, b). We next examined the distribution of HRDE-1-bound small RNAs across WAGO- and CSR-target gene bodies in wild-type and the hrde-2 mutant, and CSR-1-bound small RNAs across the same genes[74]. In wild-type animals, HRDE-1-bound small RNAs are relatively evenly distributed across WAGO-target genes, while CSR-1-bound small RNAs were enriched towards the 3′ ends of CSR-target genes, consistent with previous reports (Fig. 5c)[49,75,76]. HRDE-1-bound small RNAs in the hrde-2 mutant were enriched towards the 3′ ends of CSR-target genes, similar to CSR-1 (Fig. 5c). These data are consistent with HRDE-1 binding CSR-class small RNAs produced by the RdRP EGO-1 in the hrde-2 mutant, and not by the mutator complex, which normally produces WAGO-class 22G-RNAs, aberrantly producing small RNAs from CSR-target RNAs. Together, these results demonstrate that HRDE-2 plays a critical role in promoting HRDE-1 specificity for WAGO-class 22G-RNAs.

To define the HRDE-1 target genes in wild-type and each mutant, we generated a list of genes with complementary small RNAs at least twofold enriched in the IP compared to input, with at least 10 reads per million (RPM) in the input or IP samples, and with a DESeq2 adjusted p value of ≤0.05. Comparing these lists to known targets of the CSR and WAGO pathways[50,74], we find that, in wild-type animals, the majority (58.24%, 1233 of 2117) of HRDE-1 target genes are WAGO-target genes while only 5.57% (118 of 2117) are CSR-target genes. In contrast, in the hrde-2 mutant, the majority (78.52%, 3476 of 4427) of HRDE-1 target genes are CSR-target genes while only 2.06% (91 of 4427) are WAGO-target genes (Fig. 5d). The mut-2 mutant and mut-2; hrde-2 mutant also show similar trends, where in the mut-2 mutant, 75.54% (729 of 965) of HRDE-1 target genes are WAGO-target genes while only 8.91% (86 of 965) are CSR-target genes. In the mut-2; hrde-2 mutant, 8.98% (53 of 590) of HRDE-1 target genes are WAGO-target genes and 69.15% (408 of 590) are CSR-target genes (Supplementary Fig. 5c). Therefore, we conclude that HRDE-1 is enriched for small RNAs mapping to CSR-target genes, rather than WAGO-target genes, in a hrde-2 mutant.

We next examined HRDE-1 associated small RNAs in the simr-1 mutant and the simr-1 mut-2 double mutant. We found that, similar to the hrde-2 mutant, HRDE-1 associates with small RNAs mapping to CSR-target genes in the simr-1 mutant and the simr-1 mut-2 mutant (Supplementary Fig. 5a, d). Interestingly, compared to the hrde-2 mutant, the reduction in association with small RNAs mapping to WAGO-target genes is more modest in the simr-1 mutant, resulting in enrichment for both WAGO-class and CSR-class 22G-RNAs in HRDE-1 IPs from the simr-1 mutant. Thus, SIMR-1, like HRDE-2, is required to prevent loading of CSR-class 22G-RNAs into HRDE-1 and to promote loading of WAGO-class 22G-RNAs into HRDE-1, albeit to a lesser extent than HRDE-2.

Lastly, we examined the changes in total small RNA levels in hrde-2 and simr-1 mutants using the input samples from our HRDE-1 IP experiment. We found that both hrde-2 and simr-1 mutants show an increase in small RNAs mapping to CSR-target genes and histone genes, which was observed previously for simr-1[50] (Supplementary Fig. 5e).

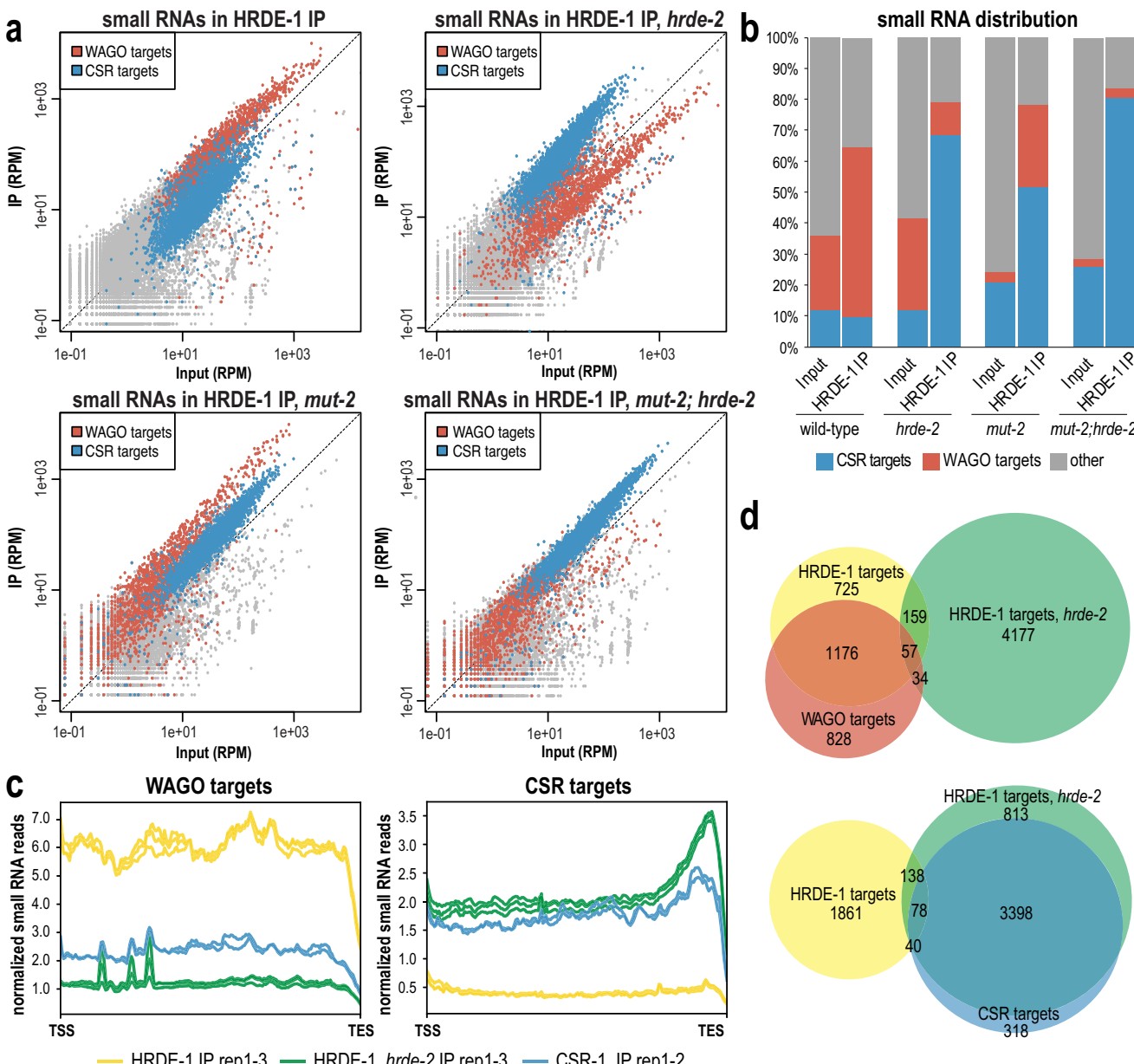

**Fig. 5 | HRDE-2 promotes specificity of HRDE-1 for WAGO-class small RNAs.**
**a** Normalized HRDE-1 IP compared to input small RNA reads for wild-type, *hrde-2*, *mut-2*, and *mut-2; hrde-2* mutants from day 1 adults. WAGO-target and CSR-target genes are indicated in red and blue, respectively. One representative replicate is shown. **b** Percentage of reads from HRDE-1 input and IP small RNA libraries mapping to CSR-target genes, WAGO-target genes, or other genes in wild-type, *hrde-2*, *mut-2*, and *mut-2; hrde-2* mutants. The percentage of reads is an average of three replicate libraries. Individual libraries are shown in Supplementary Fig. S5a. **c** Density plot of small RNA enrichment on WAGO targets and CSR targets in HRDE-1

IP (yellow), HRDE-1 IP in the *hrde-2* mutant (green), and CSR-1 IP (blue). Transcription start site (TSS) to transcription end site (TES) were plotted using normalized small RNA reads. Each of two or three replicates are represented as individual lines. **d** Venn diagrams indicate overlap of genes enriched for small RNAs in HRDE-1 IP from wild-type and *hrde-2* mutants compared to WAGO targets (top) and CSR targets (bottom). HRDE-1 target genes in wild-type and *hrde-2* mutants are defined as at least twofold enriched in the IP, with at least 10 RPM in IP samples and a DESeq2 adjusted *p* value ≤ 0.05 (see Supplementary Data 4 for gene lists).

Additionally, both *hrde-2* and *simr-1* mutants have a significant reduction in small RNAs mapping to WAGO-target and piRNA-target genes (Supplementary Fig. 5e, f). The majority of WAGO-target and piRNA-target genes overlap, but there is a subset of WAGO-targets that are piRNA and PRG-1 independent. Using a list of WAGO-target genes whose small RNAs are either unchanged or increased in *prg-1* mutants[50], we found that small RNAs mapping to these PRG-1-independent WAGO-target genes are not reduced in *hrde-2*, similar to what was previously observed for *simr-1* (Supplementary Fig. 5e)[50]. In addition, the small RNAs depleted in *hrde-2* and *simr-1* mutants largely overlap with one another and with the piRNA-target genes (Supplementary Fig. 5g).

Together, these data support that HRDE-2 and SIMR-1 function in the same pathway and downstream PRG-1, although some differences do exist in the affected small RNA targets in each mutant.

**HRDE-1-targeting is not sufficient to silence CSR-target genes**
Our data demonstrate that HRDE-1 binds to CSR-class small RNAs and localizes to the nucleus in a *hrde-2* mutant. To determine if HRDE-1 regulates CSR-target gene expression in a *hrde-2* mutant, we performed mRNA sequencing in wild-type, *hrde-2*, and *hrde-1* one-day-old adult animals. Comparing mRNA reads in *hrde-2* and *hrde-1* mutants to wild-type, we did not observe a substantial change in CSR-target gene

expression (Fig. 6a, c and Supplementary Fig. 6a), which suggests that the CSR-target genes are not silenced in a *hrde-2* mutant. We hypothesize that there might exist other mechanisms that prevent HRDE-1 from silencing those CSR-target genes co-transcriptionally, or there are antagonistic mechanisms promoting CSR-target gene expression despite HRDE-1 silencing them co-transcriptionally.

Although HRDE-1 does not silence CSR-target genes in the *hrde-2* mutant, we observed that there is a group of upregulated genes. By examining the genes in this group, we determined that these upregulated genes are spermatogenic genes and targets of the ALG-3/4 pathway (Fig. 6b, c)[17,77]. Surprisingly, the expression of spermatogenic genes in the *hrde-1* mutant was largely unchanged compared to wild-type (Fig. 6c and Supplementary Fig. 6b), which is in contrast to previous work indicating that spermatogenic genes are negatively regulated by HRDE-1[78]. This discrepancy may be due to differences in developmental staging, since we are looking at day-one adults while Cornes et al. (2022) were using L4-stage animals. Thus, together with previously published data[78], our results indicate that both HRDE-1 and HRDE-2 negatively regulate spermatogenic genes, though neither mutant substantially affects CSR- or WAGO-target gene expression.

Next, we compared changes in small RNA enrichment following HRDE-1 IP to changes in mRNA expression in a *hrde-2* mutant relative to wild-type. We found that the majority of CSR-target genes have no change in mRNA expression in a *hrde-2* mutant compared to wild-type, despite CSR-class 22G-RNAs being highly enriched in the HRDE-1-loaded small RNAs in *hrde-2* mutants. Interestingly, a handful of CSR-target genes do show both enrichment in HRDE-1 IPs and reduced mRNA expression in the *hrde-2* mutant. Similarly, the majority of WAGO-target genes have no change in mRNA expression in the *hrde-2* mutant compared to wild-type, though the WAGO-class 22G-RNAs are no longer associated with HRDE-1 in the *hrde-2* mutant. However, like for CSR-1, a subset of genes (3.4%, 67 of 1962 total WAGO targets) do show a 2-fold or greater increase in mRNA expression that correlates with a 2-fold or greater reduction in HRDE-1 small RNA loading in the *hrde-2* mutant compared to wild-type (Fig. 6d). Similarly, 13.5% of spermatogenic genes (349 of 2593 genes) have both a 2-fold or greater increase in mRNA expression and a 2-fold or greater reduction in HRDE-1 small RNA loading in the *hrde-2* mutant compared to wild-type (Fig. 6e and Supplementary Fig. 6c, d). These results indicate that a subset of WAGO-target and spermatogenic genes are normally targeted by HRDE-1 in adult animals, and when that targeting is lost in a *hrde-2* mutant, they show an increase in mRNA expression. These results are consistent with a recent publication demonstrating that piRNAs can trigger transcriptional silencing of spermatogenic genes mediated by HRDE-1[78]. While we do not observe a similar upregulation of spermatogenic genes in *hrde-1* mutants (Supplementary Fig. 6b), we hypothesize that subtle differences in developmental timing between the *hrde-1* and *hrde-2* mutants and differences in staging between our experiments and Cornes et al.[78] could lead to this disparity. It will be interesting to further investigate differences between the subset of WAGO-class and spermatogenic genes for which mRNA expression is upregulated upon loss of HRDE-1 targeting, and those for which mRNA expression is unchanged.

## HRDE-1 promotes H3K9me3 deposition at CSR targets in a *hrde-2* mutant

It has been shown previously that HRDE-1 binds WAGO-class small RNAs and recruits methyltransferases (e.g. MET-2, SET-25, SET-32) to deposit the H3K9me3 silencing signal on its target genes[35,79–81]. Here, we have shown that HRDE-1 loads CSR-class small RNAs in a *hrde-2* mutant but does not silence CSR-class genes at the mRNA level. To determine if HRDE-1 promotes H3K9me3 deposition onto CSR-class genes in a *hrde-2* mutant, we examined the levels of H3K9me3 across the genome in wild-type, *hrde-2*, and *hrde-1* mutants. We first isolated germline nuclei from one-day-old adult animals and then performed CUT&Tag sequencing

targeting the H3K9me3 modification in the isolated germline nuclei[82,83] (Fig. 7a). CUT&Tag libraries were constructed with biological triplicates for each strain, and replicates showed a high correlation with one another (Supplementary Fig. 7a). To validate the specificity of the antibody, we included the SNAP-CUTANA spike-in control (EpiCypher) which is a panel of modified DNA-barcoded designer nucleosomes. Surprisingly, we discovered that the anti-H3K9me3 antibody (abcam-ab889b), in addition to recognizing H3K9me3, demonstrates significant cross-reactivity with two other tri-methyl silencing marks, H3K27me3 and H4K20me3[84,85] (Supplementary Fig. 7b). Since this anti-H3K9me3 antibody has been widely used in previous research for ChIP-qPCR and ChIP-seq to characterize H3K9me3 methylation levels in different organisms including *C. elegans*[40–42,79,81,86–88], we continued our downstream analysis with the caveat that we may be assaying multiple silencing marks (H3K9me3, H3K27me3, and H4K20me3) simultaneously.

To further validate our libraries, we first examined the H3K9me3 levels at germline nuclear RNAi-dependent heterochromatic (GRH) loci, defined previously as regions that lose H3K9me3 in a *hrde-1* mutant using a 1.5-fold cutoff[79]. We found that the H3K9me3 levels are decreased at GRH sites in our *hrde-1* mutant libraries, indicating that our anti-H3K9me3 CUT&Tag results are consistent with previous anti-H3K9me3 ChIP-seq results (Fig. 7b). We next examined the H3K9me3 level across the whole genome and WAGO targets in both *hrde-1* and *hrde-2* mutants and found that there was no significant change in H3K9me3 deposition (Supplementary Fig. 7c, d). These results are consistent with previous studies indicating that the relationship between HRDE-1 bound small RNAs, target mRNA expression, and H3K9me3 deposition are complex[79].

Next, to determine if HRDE-1 promotes deposition of H3K9me3 at CSR-target genes in the *hrde-2* mutant, when it loaded primarily with CSR-class 22G-RNAs, we examined H3K9me3 levels at CSR-target genes and found substantially increased H3K9me3 at CSR targets in the *hrde-2* mutant compared to wild-type (Fig. 7c and Supplementary Fig. 7e). When analyzing the H3K9me3 RPM across all CSR- and WAGO-target genes, we noticed that the majority of CSR-target genes exhibited an increase in H3K9me3 in the *hrde-2* mutant, whereas WAGO-target genes show more variability, with some genes exhibiting an increase and others a decrease in H3K9me3 in the *hrde-2* mutant (Fig. 7d). Overall, CSR-target genes consistently show a significant increase in H3K9me3, whereas WAGO target genes show no significant change or a modest reduction in H3K9me3 depending on the method of analysis (Fig. 7e, f).

Furthermore, we annotated the identified peaks and obtained a list of genes exhibiting significantly increased H3K9me3 in a *hrde-2* mutant compared to wild-type. By comparing this list of genes to WAGO targets, CSR targets, and the targets of HRDE-1 in a *hrde-2* mutant, we found that the genes with increased H3K9me3 overlap more substantially with CSR-1 targets and HRDE-1 targets in the *hrde-2* mutant than WAGO targets (Supplementary Fig. 7f). H3K9me3 levels at HRDE-1 targets in the *hrde-2* mutant are also substantially increased in the *hrde-2* mutant compared to wild-type (Fig. 7g), as expected given the overlap with CSR targets. Next, to determine whether the level of HRDE-1-bound 22G-RNAs targeting CSR-class genes in a *hrde-2* mutant correlates with the levels of H3K9me3 at those loci, we divided the CSR target genes into quartiles based on their $log_2$(fold change small RNA) of HRDE-1 IP/input in the *hrde-2* mutant. We found that the quartile of CSR-target genes with the highest enrichment in the HRDE-1 IP also displayed the greatest increase in H3K9me3 deposition in the *hrde-2* mutant (Fig. 7h). Finally, to visualize the relationship between CSR-class 22G-RNA association with HRDE-1 and H3K9me3 deposition at individual gene loci, we selected some exemplary CSR-1 targets (*eif-3.I*, *ddx-10*, and Y23H5B.5) and examined the distribution of small RNA reads and H3K9me3 deposition across the genes. We found that H3K9me3 levels increased dramatically at the sites targeted by HRDE-1-bound small RNAs in a *hrde-2* mutant (Fig. 7i, Supplementary Fig. 7g).

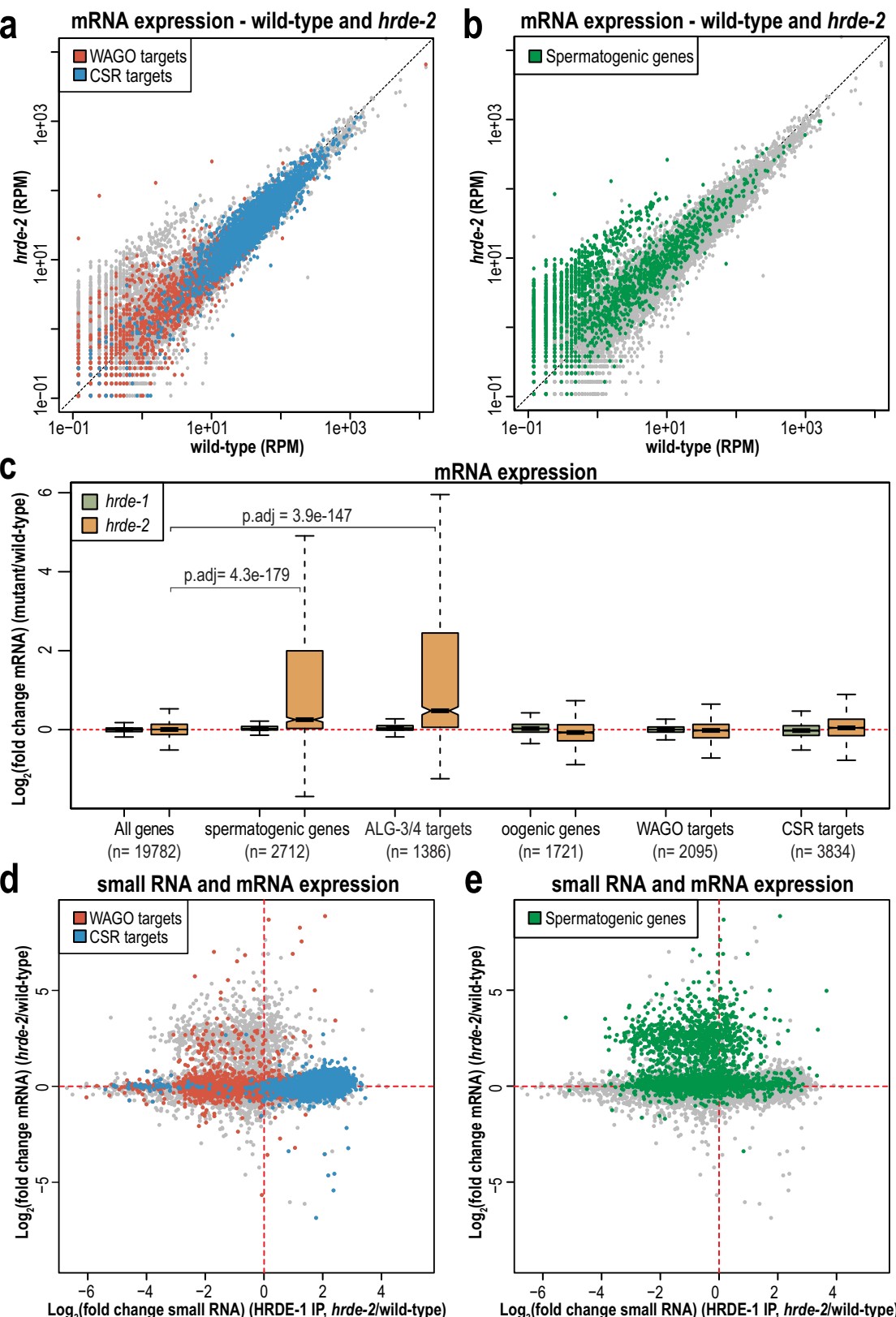

Altogether, our results demonstrate that HRDE-1 can promote deposition of H3K9me3 at CSR-target gene loci in the *hrde-2* mutant when CSR-class 22G-RNAs are loaded into HRDE-1. It will be interesting to further investigate why the loading of CSR-class 22G-RNAs into HRDE-1 is sufficient to promote H3K9me3 deposition at CSR-target genes but that this increase in H3K9me3 is not sufficient to cause a

reduction in mRNA expression. One previous study has shown that a large number of endogenous HRDE-1 target genes become transcriptionally active in a *hrde-1* mutant only when raised at elevated temperature for multiple generations[89], so it could be interesting to examine the mRNA and H3K9me3 levels at CSR targets in a *hrde-2* mutant at elevated temperature.

**Fig. 6 | Spermatogenic genes are de-silenced in a *hrde-2* mutant. a** Normalized mRNA reads in *hrde-2* mutants compared to wild-type from day 1 adults. WAGO-target and CSR-target genes are indicated in red and blue, respectively. One representative replicate is shown. **b** Normalized mRNA reads in *hrde-2* mutants compared to wild-type from day 1 adults. Spermatogenic genes are indicated in green. One representative replicate is shown. **c** Box plots depicting log2(fold change mRNA abundance) in *hrde-1* and *hrde-2* mutants compared to wild-type for three biological replicates. Bolded midline indicates median value, box indicates the first and third quartiles, and whiskers represent the most extreme data points within 1.5 times the interquartile range, excluding outliers. Two-tailed *t*-tests were

performed to determine statistical significance and *p* values were adjusted for multiple comparisons. Adjusted *p* values are a comparison of the indicated gene list to all genes in the *hrde-2* mutant. **d** Scatter plot displaying each gene as a function of its log2 fold change for the level of small RNAs immunoprecipitated by HRDE-1 (x-axis) and mRNA expression (y-axis) in *hrde-2* mutant compared to wild-type. WAGO class and CSR class genes are indicated in red and blue, respectively. **e** Scatter plot displaying each gene as a function of its log2 fold change for the level of small RNAs immunoprecipitated by HRDE-1 (x-axis) and mRNA expression (y-axis) in *hrde-2* mutant compared to wild-type. Spermatogenic genes are indicated in green.

## Discussion

How each class of small RNAs is correctly sorted into the correct Argonaute proteins, and the precise molecular function of each phase-separated germ granule domain are two major questions in the *C. elegans* small RNA field. In this study, we demonstrate that HRDE-2 localizes to SIMR foci where it recruits the Argonaute protein HRDE-1, only when HRDE-1 is not loaded with a 22G-RNA. HRDE-2 then mediates small RNA sorting, ensuring that only WAGO-class and not CSR-class 22G-RNAs are loaded into HRDE-1. We further show that unloaded HRDE-1 cannot translocate to the nucleus, even when HRDE-2 is absent (Fig. 8a, b).

How does HRDE-2 mediate small RNA sorting for HRDE-1? We have previously shown that *Mutator* foci are comprised of proteins that promote WAGO-class 22G-RNA production and we proposed that they constitute a siRNA amplification compartment[33]. SIMR foci and *Mutator* foci are exceptionally close to one another, and SIMR-1 was first identified as a mass spectrometry interactor of MUT-16[50]. Thus, *Mutator* foci, and by proximity SIMR foci, may contain a high concentration of unbound WAGO-class 22G-RNAs, making them the ideal sites to recruit unloaded HRDE-1. In fact, loss of germ granules leads to a significant reduction in WAGO-class 22G-RNAs[76]. In contrast, the same loss of germ granules does not affect CSR-class 22G-RNA biogenesis, suggesting that CSR-class 22G-RNAs are not produced in germ granules and may rather be produced in the cytosol[76]. These data lead us to the hypothesis that HRDE-2 provides small RNA specificity to HRDE-1 by recruiting unloaded HRDE-1 to the vicinity of unbound WAGO-class 22G-RNAs, and in a *hrde-2* mutant, where unloaded HRDE-1 is not recruited to germ granules, HRDE-1 instead binds CSR-class 22G-RNAs which are in high concentration in the cytosol (Fig. 8a, b, Supplementary Fig. 8a, b). It remains possible, however, that germ granule localization is not essential to promote proper HRDE-1 loading, but rather that HRDE-2 can promote correct small RNA loading of HRDE-1 both in the germ granule and diffusely in the cytoplasm. Further investigation into the molecular function of HRDE-2 will help to shed light on the requirement of granule localization for small RNA loading specificity. Regardless of the class of 22G-RNAs, once loaded, HRDE-1 translocates to the nucleus where it mediates transcriptional silencing.

How is loaded HRDE-1 translocated to the nucleus? In this work we have shown that unloaded HRDE-1 is completely restricted to the cytosol, similar to HRDE-1 with a mutation in the NLS. The NLS of HRDE-1 is located in the unstructured N-terminal extension of the protein and is critical for nuclear import of HRDE-1. Flexible N-terminal extensions are found in many other Argonaute proteins and have, in some cases, been shown to be important for nuclear import and export. For example, an N-terminal truncation of the *Drosophila* Piwi protein, which contains an NLS, leads to loss of nuclear accumulation[90], while the nucleo-cytosolic shuttling of *Arabidopsis* AGO1 is regulated by both an N-terminally-located nuclear export signal (NES) and NLS that may be differentially utilized depending on small RNA-binding state[91]. Formation of the mature Argonaute/small RNA complex also appears to be key for nuclear import of Argonautes from a wide range of organisms. In *Tetrahymena*, the Argonaute protein Twi1p is selectively transported to the nucleus only when bound to its siRNA partners, and dependent on the

Giw1p protein[92]. Similarly, *Arabidopsis* AGO4 requires the presence of siRNAs for nuclear import[93]. Multiple studies have suggested a model where the NLS of a nuclear Argonaute protein is hidden when the protein is not bound to small RNAs, either through intramolecular folding or by binding of an interacting protein[91,93,94]. Thus, we hypothesize that HRDE-1, as well as its somatic counterpart NRDE-3, which also requires an NLS and bounds siRNAs for nuclear import[36], may reveal the NLS only upon siRNA loading, though further investigation will be needed to test this possibility.

This transition could also be regulated by HRDE-1 phosphorylation, as HRDE-1 is predicted to have a phosphorylation site at Y669 in the MID domain[95]. This site is homologous to the Y529 site in human Ago2, which, when phosphorylated, inhibits miRNA binding[96]. Likewise, phosphorylation of the MID domain of *C. elegans* miRNA Argonaute ALG-1 impairs its ability to associate with miRNAs[97]. Many Argonaute phosphorylation sites are conserved between species, including human, mouse, rat, zebrafish, and *C. elegans*[98]. Thus, it will be interesting to identify *bone fide* HRDE-1 phosphorylation sites by mass spectrometry and explore how phosphorylation affects 22G-RNA binding, interaction with partners such as HRDE-2, as well as localization to the nucleus and SIMR foci.

We also observed a significant reduction in germ granule association for the HRDE-1 NLS mutant and the IDR deletion. This reduction could suggest that the HRDE-1 NLS is also important for the interaction between HRDE-1 and HRDE-2. However, since we still observe HRDE-1 associating with germ granules when both the NLS and small RNA binding are disrupted (Supplementary Fig. 3e), it is more likely that HRDE-1 only associates with HRDE-2 in its small RNA-unbound state. Accordingly, small RNAs are likely to be loaded normally in the NLS and IDR mutants, resulting in the loss of germ granule association and redistribution of the small RNA-loaded HRDE-1 protein into the cytosol.

Is promoting HRDE-1 small RNA specificity the primary function of SIMR foci? In previous work, we characterized the SIMR-1 protein and found that it was required for the production of 22G-RNAs at piRNA-target loci[50]. Here we find that the *hrde-2* mutant has very similar small RNA defects, including both a reduction in 22G-RNAs at piRNA-target loci and an increase in small RNAs mapping to histone genes (Supplementary Fig. 5e–g). Further, there are many parallels between the *simr-1* and *hrde-2* loss of function phenotypes – both have mortal germline phenotypes at elevated temperature that can be rescued by returning the animals to permissive temperature and both can desilence a piRNA sensor[50,66]. However, there are some differences between the *simr-1* and *hrde-2* mutant phenotypes. With regard to the mortal germline phenotype, a *hrde-2* mutant is sterile within 4 generations at elevated temperature while a *simr-1* mutant is not completely sterile until generation 11 at elevated temperature. Consistent with a *simr-1* mutant having generally milder phenotypes compared to *hrde-2*, in this work we found that HRDE-2 is essential to recruit unloaded HRDE-1 to germ granules while SIMR-1 is only partially required (Fig. 3a, d and Supplementary Fig. 3a). Similarly, in a *hrde-2* mutant, HRDE-1 targets only CSR-class genes while, in a *simr-1* mutant, HRDE-1 targets both WAGO-class and CSR-class genes (Fig. 5a and Supplementary Fig. 5d). There are also some differences in localization

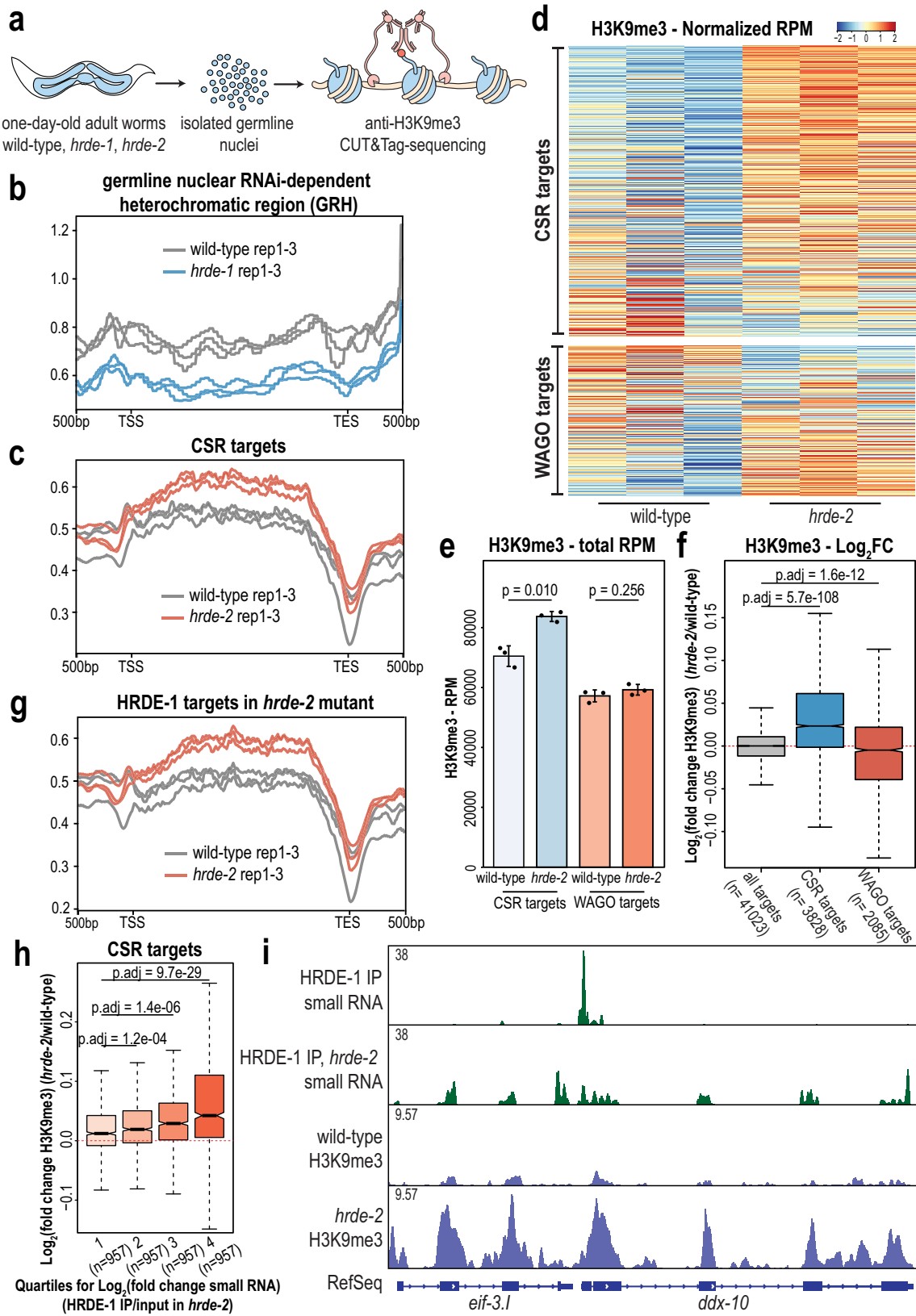

between SIMR-1 and HRDE-2, where HRDE-2 shows an increase in expression during spermatogenesis[67], while SIMR-1 is expressed more uniformly throughout the germline and across development. Despite these differences, the similarities between *simr-1* and *hrde-2* mutants suggest both play a role in silencing piRNA targets, as does HRDE-1[35,78]. Thus, it is possible that all phenotypes associated with *simr-1* and

*hrde-2* mutants could be attributed to the loss of HRDE-1 targeting to WAGO-target genes and the mistargeting of HRDE-1 to CSR-target genes. Additionally, we have not investigated CSR-1 small RNA loading in *simr-1* or *hrde-2* mutants, so it is possible that some *simr-1* and *hrde-2* phenotypes result from reduced CSR-1 function stemming from the competition between CSR-1 and HRDE-1 for CSR-class 22G-RNAs.

**Fig. 7 | H3K9me3 is enriched at CSR-1 target genes in *hrde-2* mutants.**
**a** Schematic representation of germline nuclei isolation followed by CUT&Tag sequencing. **b** Density plot of H3K9me3 level on germline nuclear RNAi-dependent heterochromatic region (GRH) with one point five log fold threshold in wild-type (gray) and *hrde-1* mutants (blue). **c** Density plot of H3K9me3 level on CSR-target genes in wild-type (gray) and *hrde-2* mutants (red). **d** Heatmap of CSR targets (top) and WAGO targets (bottom) normalized H3K9me3 RPM reads in wild-type (left three lanes) and *hrde-2* mutants (right three lanes). **e** Total RPM of CSR targets (blue) and WAGO targets (red) in wild-type and *hrde-2* mutants. Bar graphs representing mean for three biological replicates and error bars indicate Standard Deviation (SD). Two-tailed *t*-tests were performed to determine statistical significance. **f** $\log_2$(fold change H3K9me3) of all targets (gray), CSR targets (blue), and WAGO targets (red) in wild-type and *hrde-2* mutants. **g** Density plot of H3K9me3 level on WAGO-target genes in wild-type (gray) and *hrde-2* mutants (red).
**h** $\log_2$(fold change) of H3K9me3 at CSR targets, sorted into 4 quartiles based on

the $\log_2$(fold change) of small RNAs(HRDE-1 IP/input in *hrde-2*). The first quartile represents CSR targets with the lowest HRDE-1 IP/input small RNA $\log_2$(fold change), and the fourth quartile includes CSR targets with the highest HRDE-1 IP/input small RNA $\log_2$(fold change). **i** Normalized small RNA and H3K9me3 read distribution across two CSR target genes (*eif-3.I* and *ddx-10*) in wild-type and *hrde-2* mutants. For all box plots (**f, h**), bolded midline indicates median value, box indicates the first and third quartiles, and whiskers represent the most extreme data points within 1.5 times the interquartile range, excluding outliers. Two-tailed *t*-tests were performed to determine statistical significance and *p* values were adjusted for multiple comparisons. For all density plots (**b, c, g**), 500 bp upstream transcription start site (TSS) and downstream transcription end site (TES) were plotted using normalized H3K9me3 reads. For density plots and heat map (**b–d, g**) all three replicates are shown. For normalized small RNA reads (**i**), one representative replicate is shown.

Do all Argonaute proteins have a loading site to promote specificity? In this work we have shown that HRDE-2 recruits unloaded HRDE-1 to SIMR foci to promote correct small RNA loading. We think it is unlikely that HRDE-2 is required for small RNA loading for other germline-expressed Argonaute proteins based on our HRDE-2 IP-mass spec data identifying only one Argonaute protein, HRDE-1, and evidence from Spracklin et al. indicating that a mutation in *hrde-2* does not affect the level of exogenous *oma-1* siRNAs loaded into PPW-2/WAGO-3[66]. Furthermore, a recent publication identified two paralogs of HRDE-2, ENRI-1 and ENRI-2, which immunoprecipitate with the somatic nuclear Argonaute protein, NRDE-3, in its unloaded state and are thought to play a role in preventing misloading of small RNAs into NRDE-3 at inappropriate times or locations[67]. Thus, it seems likely that the HRDE-2/ENRI proteins are acting in parallel to promote correct small RNA loading for the two nuclear Argonaute proteins, HRDE-1 and NRDE-3, though it is not yet known whether NRDE-3 is similarly recruited to a specific subcellular location for this purpose. Yet there are 19 functional Argonaute proteins in *C. elegans*, and it is still unclear why only the two nuclear Argonaute proteins require HRDE-2/ENRI proteins for specificity. Further work will be necessary to determine how the remaining Argonaute proteins derive their small RNA specificity, and whether, like HRDE-2, their subcellular or germ granule localization plays an important role in dictating small RNA loading.

Here we show that HRDE-2 and SIMR-1 recruit unloaded HRDE-1 to germ granules to promote WAGO-class 22G-RNA loading and simultaneously prevent CSR-class 22G-RNA loading. Our results provide a framework for understanding one mechanism by which small RNA binding specificity is achieved for an Argonaute protein. While some of the 19 functional *C. elegans* Argonaute proteins may have unique tissue-specific, temporal, or spatial expression patterns, many have overlapping expression and localization and still bind unique complements of small RNAs. To our knowledge, only NRDE-3 has so far been shown to have distinct localization patterns when bound or unbound to 22G-RNAs[36]. Thus, our work opens up new questions and new avenues of study for determining how each Argonaute protein recognizes and binds to the correct small RNA partners.

## Methods

### C. elegans strains
*C. elegans* strains were maintained at 20 °C on NGM plates seeded with OP50 *E. coli* according to standard conditions unless otherwise stated[99]. All strains used in this project are listed in Supplementary Data 2.

### CRISPR-mediated strain construction
For *mut-2(cmp307), mut-2(cmp308), hrde-1(cmp309[HK-AA]), mut-16(cmp313[mut-16::3xmyc]), mut-16(cmp314[mut-16::3xmyc]), hrde-1(cmp315[KRS-AAA]), hrde-2(cmp316[HELICΔ]),* and *hrde-1(cmp322[IDRΔ]),*

we used an oligo repair template and RNA guide (Supplementary Data 3). For injections using a single gene-specific crRNA, the injection mix included 0.25 µg/µl Cas9 protein (IDT), 100 ng/µl tracrRNA (IDT), 14 ng/µl dpy-10 crRNA, 42 ng/µl gene-specific crRNA, and 110 ng/µl of the oligo repair template. For injections using two gene-specific crRNAs, the injection mix included 0.25 µg/µl Cas9 protein (IDT), 100 ng/µl tracrRNA (IDT), 14 ng/µl dpy-10 crRNA, 21 ng/µl each gene-specific crRNA, and 110 ng/µl of each repair template. The following strains were used: *mut-2(cmp307)* into USC1476: *mut-16(cmp41[mut-16::mCherry::2xHA])* I; *hrde-1(tor125[GFP::3xFLAG::hrde-1])* III, *mut-2(cmp308)* into USC1477: *simr-1(cmp15[simr-1::mCherry::2xHA])* I; *hrde-1(tor125[GFP::3x-FLAG::hrde-1])* III, *hrde-1(cmp309[HK-AA]), hrde-1(cmp315[KRS-AAA]), and hrde-1(cmp322[IDRΔ])* into JMC231: *hrde-1(tor125[GFP::3xFLAG::hrde-1])* III, *hrde-1(cmp340[HK-AA]* into USC1510: *hrde-1(tor125[GFP::3xFLAG::hrde-1], cmp315[KRS-AAA])* III, *mut-16(cmp313[mut-16::3xMyc])* into USC1455 *simr-1(cmp15[simr-1::mCherry::2xHA])* I; *hrde-2/enri-3(tag1660[hrde-2/enri-3::2xTy1::GFP])* V, *mut-16(cmp314[mut-16::3xMyc])* into USC1479: *simr-1(cmp15[simr-1::mCherry::2xHA]) mut-2(cmp308)* I; *hrde-1(tor125[GFP::3xFLAG::hrde-1])* III, and *hrde-2(cmp316[HELICΔ])* into TRG1656: *hrde-2/enri-3(tag1660[hrde-2/enri-3::2xTy1::GFP])* V. Following injection, F1 animals with the Rol phenotype were isolated and genotyped by PCR to identify heterozygous animals with the mutations of interest, then F2 animals were further singled out to identify homozygous mutant animals.

### Live imaging and immunofluorescence imaging
Live imaging of *C. elegans* was performed in M9 buffer containing sodium azide to prevent movement. For immunofluorescence, *C. elegans* were dissected in egg buffer containing 0.1% Tween-20 and fixed in 1% formaldehyde in egg buffer as described[100]. Samples were immunostained with anti-FLAG 1:500 (Sigma Aldrich, F1804), anti-PGL-1 1:100 (DSHB K76), anti-HA 1:500 (Roche, 11867423001), anti-Myc 1:100 (Thermo Fisher, 13-2500), and anti-GFP 1:500 (Thermo Fisher, A-11122). Alexa-Fluor secondary antibodies were purchased from Thermo Fisher. Anti-mouse IgG Alexa Fluor 488 1:1000 (Thermo Fisher A11029), anti-Rat IgG Alexa Fluor 555 1:1000 (Thermo Fisher A21434), anti-mouse IgM Alexa Fluor 647 1:500 (Thermo Fisher A21238), and anti-mouse IgG Alexa Fluor 647 1:500 (Thermo Fisher A21236) were used. Animals were dissected at the adult stage (72 h post synchronized L1, or 24 h post L4). Imaging was performed on a DeltaVision Elite microscope (GE Healthcare) using a 60x N.A. 1.42 oil-immersion objective, except for Figs. 1d and 3e. For immunofluorescence images, data stacks were collected and deconvolution was performed followed by generation of three-dimensional images by maximum intensity projection using the SoftWoRx package. Confocal imaging for Figs. 1d and 3e was performed on a Leica Stellaris 5 confocal microscope using a 63x NA 1.40 oil-immersion objective and the Lightening deconvolution package. All images were pseudocolored and adjusted for brightness/contrast using Adobe Photoshop.

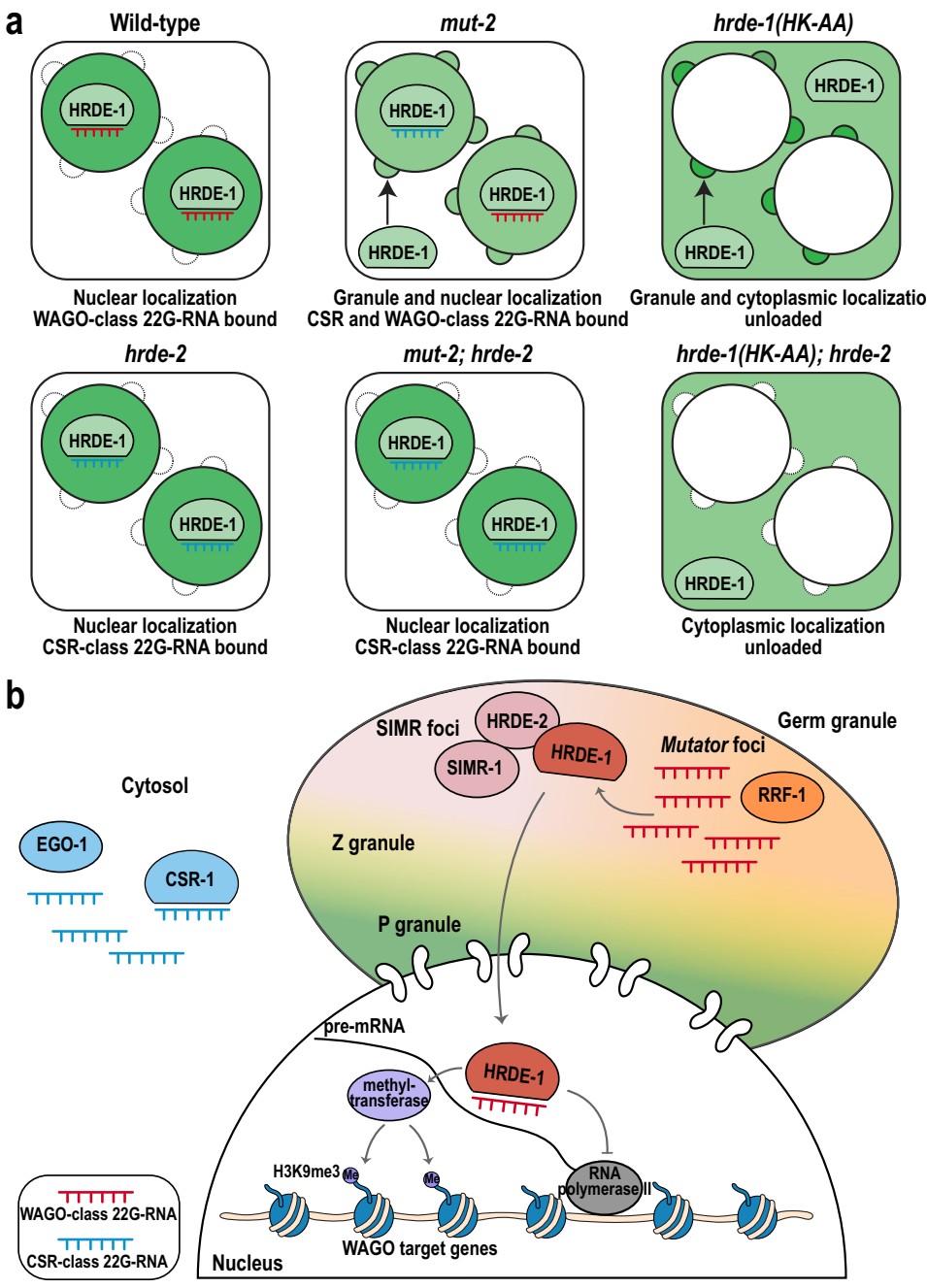

**Fig. 8 | Model: HRDE-2 promotes germ granule localization and small RNA binding for HRDE-1. a** HRDE-1 localization (green) and class of associated small RNAs are depicted for wild-type and various mutants. In wild-type, HRDE-1 associates with WAGO-class 22G-RNAs and localizes to the nucleus. In the *hrde-2* mutant, HRDE-1 associates with CSR-class 22G-RNAs while still localizing to the nucleus. In the *mut-2* mutant, HRDE-1 associates with both WAGO and CSR-class 22G-RNAs and localizes to both nucleus and germ granule. In the *mut-2; hrde-2* mutant, HRDE-1 associates with CSR-class 22G-RNAs and localizes to the nucleus. In *hrde-1(HK-AA)* mutant, HRDE-1 remains unloaded, is excluded from the nucleus, and localizes to germ granules and the cytoplasm. In the *hrde-1(HK-AA); hrde-2* mutant, HRDE-1 is unloaded and dispersed in the cytoplasm. **b** HRDE-2 is a component of SIMR foci, where it interacts with SIMR-1 and recruits unloaded HRDE-1. In SIMR foci, HRDE-1 loads WAGO-class 22G-RNAs, which are produced in adjacent *Mutator* foci by the RNA-dependent RNA polymerase RRF-1. Upon 22G-RNA loading, HRDE-1 is translocated to the nucleus, recognizes pre-mRNA complementary to the bound 22G-RNA, and recruits downstream NRDE pathway factors and methyltransferase to deposit the H3K9me3 silencing marker on HRDE-1 target genes and inhibit RNA polymerase II transcription. In contrast, CSR-class 22G-RNAs are produced in the cytosol by RNA-dependent RNA polymerase EGO-1, resulting in spatial separation from WAGO-class 22G-RNAs.

## Granule distance quantification

Quantification of distance between foci centers was performed in FIJI/ImageJ2 (version 2.9.0) according to published methods[48]. We imaged pachytene germ cell nuclei from three animals. Two granules were selected from each of four germ cells for a total of 24 granules per animal. Z stacks were opened using the 3D object counter plugin for ImageJ to collect the $x$, $y$, and $z$ coordinates for the center of each desired foci[101]. With these coordinates, distances between the foci centers were calculated using the formula $\sqrt{(x2-x1)^2+(y2-y1)^2+(z2-z1)^2}$. To account for chromatic shift between channels, distances were calculated between each pair of channels using TransFluorospheres streptavidin-labeled microspheres, 0.04 μm

(Thermo Fisher, T10711) and these distances were used to correct granule distances.

## Yeast two-hybrid analysis

Yeast two-hybrid screening was performed by Hybrigenics Services, S.A.S., Evry, France (http://www.hybrigenics-services.com). The coding sequence for full-length *simr-1* (NM_001380434.1) was PCR-amplified and cloned into pB29 as an N-terminal fusion to LexA (simr-1-LexA). The construct was checked by sequencing the entire insert and used as a bait to screen a random-primed *C. elegans* Mixed Stages cDNA library constructed into pP6. pB29 and pP6 derive from the original pBTM116[102,103] and pGADGH[104] plasmids, respectively. 69 million clones (7-fold the complexity of the library) were screened using a mating approach with YHGX13 (Y187 ade2-101::loxP-kanMX-loxP, mata) and L40DGal4 (mata) yeast strains as previously described[105]. 180 His+ colonies were selected on a medium lacking tryptophan, leucine and histidine, and supplemented with 5 mM 3-aminotriazole to handle bait autoactivation. The prey fragments of the positive clones were amplified by PCR and sequenced at their 5′ and 3′ junctions. The resulting sequences were used to identify the corresponding interacting proteins in the GenBank database (NCBI) using a fully automated procedure. A confidence score (PBS, for Predicted Biological Score) was attributed to each interaction as previously described[106].

Further description of the confidence score: The PBS relies on two different levels of analysis. Firstly, a local score takes into account the redundancy and independency of prey fragments, as well as the distribution of reading frames and stop codons in overlapping fragments. Secondly, a global score takes into account the interactions found in all the screens performed at Hybrigenics using the same library. This global score represents the probability of an interaction being nonspecific. For practical use, the scores were divided into four categories, from A (highest confidence) to D (lowest confidence). A fifth category (E) specifically flags interactions involving highly connected prey domains previously found several times in screens performed on libraries derived from the same organism. Finally, several of these highly connected domains have been confirmed as false-positives of the technique and are now tagged as F. The PBS scores have been shown to positively correlate with the biological significance of interactions[107,108]. The protein list can be found in Supplementary Data 1.

## Western blots

Synchronized adult *C. elegans* were harvested (-72 h at 20 °C after L1 arrest) and 200 adults were loaded per lane. Proteins were resolved on 4–12% Bis-Tris polyacrylamide gels (Thermo Fisher, NW04122BOX), transferred to nitrocellulose membranes (Thermo Fisher, LC2001), and probed with rat anti-HA-peroxidase 1:1000 (Roche 12013819001), mouse anti-FLAG 1:1000 (Sigma, F1804), rabbit Anti-GFP 1: 500 (Thermo Fisher, A-11122), mouse anti-TY-1 1:2000 (Thermo Fisher, MA5-23513), or mouse anti-actin 1:10,000 (Abcam ab3280), and HRP-labeled anti-mouse IgG Secondary 1:10000 (Thermo Fisher A16078) and anti-rabbit Secondary 1:10000 (Thermo Fisher A16110). Unedited western blots provided in the Source Data File. Western Blot quantification was carried out using ImageJ (version 1.53a).

## Immunoprecipitation and mass spectrometry

For immunoprecipitation experiments followed by mass spectrometry, -200,000 synchronized adult animals (-72 h at 20 °C after L1 arrest), in biological triplicates, were washed off plates with H$_2$O and collected in IP Buffer (50 mM Tris-Cl pH 7.5, 100 mM KCl, 2.5 mM MgCl2, 0.1% Nonidet P40 substitute) containing Protease Inhibitor (Thermo Fisher A32965), frozen in liquid nitrogen, and homogenized using a mortar and pestle. After further dilution into IP buffer (1:10 packed worms:buffer), insoluble particulate was removed by centrifugation and 10% of sample was taken as "input." The remaining

lysate was used for the immunoprecipitation. Immunoprecipitation was performed at 4 °C for 1 h with pre-conjugated anti-HA affinity matrix (Roche 11815016001) or Anti-GFP Agarose (RQ2) (MBL D153-8), then washed at least 10 mins for three times in immunoprecipitation buffer. A fraction of each sample was analyzed by SDS-PAGE and silver stain to confirm efficacy of immunoprecipitation. The remaining beads were submitted to the UCLA Proteome Research Center for analysis. The protein list can be found in Supplementary Data 1.

Co-immunoprecipitation experiments used the same method described above, but with -3500 synchronized adult animals (-72 h at 20 °C following L1 arrest). Following immunoprecipitation, input and IP samples were analyzed by Western Blot to detect proteins of interest.

## Small RNA library preparation and sequencing

For immunoprecipitation followed by small RNA sequencing, -100,000 synchronized adult animals (-68 h at 20 °C after L1 arrest) were washed off plates with H$_2$O and collected in IP buffer in biological triplicates. Immunoprecipitation was performed as described above using anti-FLAG Affinity Matrix (Sigma Aldrich, A2220). HRDE-1-bound RNAs were isolated using TRIzol reagent (Thermo Fisher, 15596018), followed by chloroform extraction and isopropanol precipitation. Small RNAs (18 to 30-nt) were size selected on denaturing 15% polyacrylamide gels (Bio-Rad 3450091) from total RNA samples. Small RNAs were treated with 5′ RNA polyphosphatase (Epicenter RP8092H) and ligated to 3′ pre-adenylated adapters with Truncated T4 RNA ligase (NEB M0373L). Small RNAs were then hybridized to the reverse transcription primer, ligated to the 5′ adapter with T4 RNA ligase (NEB M0204L), and reverse transcribed with Superscript III (Thermo Fisher 18080-051). Small RNA libraries were amplified using Q5 High-Fidelity DNA polymerase (NEB M0491L) and size selected on a 10% polyacrylamide gel (Bio-Rad 3450051). Library concentration was determined using the Qubit 1X dsDNA HS Assay kit (Thermo Fisher Q33231) and quality was assessed using the Agilent BioAnalyzer. Libraries were sequenced on the Illumina NextSeq500 (SE 75-bp reads) platform. Primer sequences are available in Supplementary Data 3. Differentially expressed gene lists can be found in Supplementary Data 4. Sequencing library statistics summary can be found in Supplementary Data 5.

## Small RNA binding assay and quantification

Small RNA binding assay was performed on unamplified cDNA samples generated through small RNA library preparation. A range of cycle numbers was used to amplify the small RNA amplicons using Q5 High-Fidelity DNA polymerase (NEB M0491L). Amplified cDNA samples were loaded on 10% polyacrylamide gel (Bio-Rad 3450051), stained with SYBR Gold Nucleic Acid Gel Stain (Life Technologies, S11494) and imaged. The relative quantity of bound small RNAs were determined using ImageJ (version 1.53a).

The quantification of wild-type HRDE-1 and HRDE-1(HK-AA) in Fig. 3c was performed across a range of 8 to 14 PCR cycles (8X, 10X, 12X, 14X). A comparison at 14X revealed that wild-type HRDE-1 loads 20.9 times more small RNA than HRDE-1(HK-AA). HRDE-1(HK-AA) binds 2.28 times less small RNA than wild-type after normalizing to wild-type HRDE-1 and HRDE-1(HK-AA) protein levels. In Fig. 4b, the quantification was conducted across a range of 10 to 16 PCR cycles (10X, 12X, 14X, 16X) for wild-type HRDE-1, HRDE-1, *mut-2*, and HRDE-1(HK-AA). All samples were normalized to wild-type HRDE-1 at 10 cycles. Two biological replicates were used for the quantification.

## mRNA library preparation and sequencing

50,000 synchronized adult animals (-68 h at 20 °C after L1 arrest) were collected, in biological triplicates, by washing worms off plates using water and allowing the animals to settle on ice to generate a compact pellet. Worm pellets were resuspended in 1 ml TRIzol reagent (Thermo Fisher 15596018) and freeze-thawed on dry ice followed by vortexing.

RNA isolation was performed by chloroform extraction and iso-propanol precipitation. RNA samples were normalized to 7.5 μg per 10 μL, followed by polyA selection using the Dynabead mRNA Purification Kit (Invitrogen 61006). 100 ng of polyA selected mRNA samples were treated using NEBNext Ultra II Directional RNA Library Prep Kit for Illumina (NEB E7760S) according to the manual, using NEBNext multiplex oligos for Illumina (NEB E7335S). mRNA libraries were sent to USC sequencing core for quality control (Agilent BioAnalyzer Chip) and sequenced on the Illumina NextSeq2000 100-cycle (SE 75-bp reads) platform. Primer sequences are available in Supplementary Data 3. Differentially expressed gene lists can be found in Supplementary Data 6. Sequencing library statistics summary can be found in Supplementary Data 5.

### Germline nuclei isolation followed by CUT&Tag library preparation and sequencing

50,000 synchronized adult animals (~68 h at 20 °C after L1) were collected, in biological triplicates, and germline nuclei were isolated as previously described[82]. Worms were washed off plates and lightly crosslinked in 0.1% formaldehyde (ThermoFisher 28908) for 2 minutes, then quenched in a 1 M Tris (pH 7.5) wash. Next, worms were washed in M9 media and pre-chilled Nuclei Purification Buffer+ (50 mM HEPES pH 7.5, 40 mM NaCl, 90 mM KCl, 2 mM EDTA, 0.5 mM EGTA, 0.1% Tween-20, 0.5 mM PMSF, 0.2 mM DTT, 0.5 mM spermidine, 0.25 mM spermine, cOmplete protease inhibitor cocktail (Millipore Sigma 11873580001)). Worms were resuspended in 7 ml pre-chilled Nuclei Purification Buffer+ and dounced on ice in a Wheaton dura-grind stainless steel dounce tissue grinder (VWR 62400-675) for 12 strokes. After every 5 strokes, the samples were incubated on ice for 5 min. After grinding, samples were vortexed for 30 s then incubated on ice for 5 min to release the germline nuclei twice. The nuclei were passed through six 40 μm cell strainers (Fisherbrand 22-363-547) then passed through four 20 μm cell strainers (Pluriselect NC1004201) to remove worm debris. Isolated nuclei were pelleted at 4100 rpm at 4 °C for 4 min and resuspended in Nuclei Purification Buffer + , transferred to a nonstick 2 ml tube (Ambion AM12475) and an aliquot was DAPI-stained and counted to calculate the number of nuclei extracted using Hausser Scientific hemacytometer (VWR 15170-263). The remainder of the nuclei were pelleted, the supernatant was removed, and the nuclei were resuspended in NE1 buffer (1 ml 1 M HEPES-KOH pH 7.9, 500 μL 1 M KCl, 25 μL 1 M spermidine, 500 μL 10% Triton X-100, and 10 ml glycerol in 38 ml dH2O, 1 Roche Complete Protease Inhibitor EDTA-Free (Millipore Sigma 11873580001)), followed by slow freezing and storage at −80 °C.

CUT&Tag-sequencing was performed following the protocol[83]. 50,000 isolated frozen *C. elegans* germline nuclei were thawed at 37 °C and resuspended in 50 ml NE1 buffer, ConA beads were washed in binding buffer (200 μL 1 M HEPES pH 7.9, 100 μL 1 M KCl, 10 μL 1 M CaCl2 and 10 μL 1 M MnCl2, in 10 mL final volume of dH2O). 10ul ConA beads were added into 50,000 nuclei for each sample. anti-H3K27me3 antibody (Cell Signaling 9733 T) was used at 1:50 for positive control, no primary antibody was used for negative control. anti-H3K9me3 antibody (Abcam ab8898) were used at 1:100 for the experiment. Samples were incubated at 4 °C overnight. Anti-Rabbit IgG (Antibodies-online ABIN101961) antibody was added 1:100 to samples as the secondary antibody and incubated at room temperature for 30 mins. 2ul pA-Tn5 (EpiCypher 15-1017) was added to each sample and tagmentation was performed. DNA was isolated using Phenol:Chloroform:Isoamyl Alcohol (ThermoFisher 15593031) and chloroform. After PCR amplification using NEBNext High-Fidelity 2 × PCR Master Mix (VWR 102500-096) and post-PCR cleanup using SPRI beads (Beckman Coulter A63880), the samples were sent to USC sequencing core for quality control (Agilent BioAnalyzer Chip) and sequenced on the Illumina NextSeq2000 100-cycle (PE 50-bp reads) platform. Primer sequences are available in Supplementary Data 3.

Differentially expressed gene lists can be found in Supplementary Data 7. Sequencing library statistics summary can be found in Supplementary Data 5.

### Bioinformatic analysis

For small RNA libraries, sequences were parsed from adapters and quality filtered using FASTX-Toolkit (version 0.0.13)[109]. Contamination from reads mapping to 18-mer and 28-mer size standards was filtered out using Cutadapt (version 3.4)[110]. Filtered reads were mapped to the *C. elegans* genome, WS258, using Bowtie2 (version 2.5.0)[111]. Mapped reads were assigned to genomic features using featureCounts which is part of the Subread package (version 2.0.1)[112]. Differential expression analysis was performed using Deseq2 (version 1.38.3)[113]. Deeptools (version 3.5.1) was used to generate metagene plots with BPM (per bin) normalization parameter (Bins Per Million = number of reads per bin/ sum of all reads per bin in millions)[114]. To define gene lists from IP experiments, a twofold-change cutoff, a DESeq2 adjusted *p* value of ≤0.05, and at least 10 RPM in the IP libraries were required to identify genes with significant changes in small RNA levels. The all siRNA targets list (6,582 targets) in Supplementary Fig. 5e was defined as genes having at least 10 RPM in wild-type, *hrde-2*, or *simr-1* mutant small RNA libraries, and excluding all miRNAs and piRNAs.

For mRNA libraries, adapter sequences were trimmed using Trimmomatic (version 0.39)[115] and mapped to the *C. elegans* genome, WS258, using HISAT2 (version 2.2.1)[116]. Mapped reads for genomic features were counted using featureCounts which is part of the Subread package (version 2.0.1)[112]. Differential expression analysis was performed using Deseq2 (version 1.38.3)[113]. The all genes list (19,782 genes) in Fig. 6c was defined as genes having RPM > 0 in wild-type, *hrde-1*, or *hrde-2* mutant mRNA libraries, and excluding all miRNAs and piRNAs.

For CUT&Tag libraries, reads were mapped to the *C. elegans* genome, WS258, using bowtie2 (version 2.5.0)[111]. Mapped reads for genomic features were counted using featureCounts which is part of the Subread package (version 2.0.1)[112]. Bedtools (version 2.30.0) was used to generate the density plots[117]. IGV (version 2.12.0) was used to visualize the H3K9me3 and small RNA level[118]. SEACR (version 1.3) was used to call peaks[119]. DiffBind (version 3.17) was used for differential expression analysis[120]. Deeptools (version 3.5.1) was used to generate metagene plots with BPM (per bin) normalization parameter (Bins Per Million = number of reads per bin/sum of all reads per bin in millions)[114]. The all genes list (41,023 genes) in Fig. 7f was defined as genes having RPM > 0 in wild-type or *hrde-2* mutant CUT&Tag libraries, including all miRNAs and piRNAs.

Venn diagrams were generated using BioVenn[121]. CSR-target genes, WAGO-target genes, spermatogenesis-enriched genes, and oogenesis-enriched genes were previously described[50,74,77]. Sequencing data is summarized in the Supplemental Data File 7.

### Reporting summary

Further information on research design is available in the Nature Portfolio Reporting Summary linked to this article.

## Data availability

The RNA sequencing and CUT&Tag sequencing data generated in this study are available through Gene Expression Omnibus (GEO) under accession code GSE239291. The mass spectrometry proteomics data generated in this study are available in the MassIVE repository with the dataset identifier MSV000092546. Source data are provided with this paper.

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

## Acknowledgements

We thank the members of the Phillips lab for helpful discussions and feedback on the manuscript, and the labs of Julie Claycomb, Thomas Duchaine, and Mihail Sarov for generously providing strains. This work was supported by the National Institute of Health grant R35 GM119656 (to CMP). CMP is a Pew Scholar in the Biomedical Sciences supported by the Pew Charitable Trusts (www.pewtrusts.org). Some strains were provided by the CGC, which is funded by NIH Office of Research Infrastructure Programs (P40 OD010440). Next generation sequencing was performed by the USC Molecular Genomics Core, which is supported by award number P30 CA014089 from the National Cancer Institute and mass spectrometry was performed by the UCLA Proteome Research Center.

## Author contributions

S.C.: Conceptualization, Investigation, Formal analysis, Writing–original draft, Writing–reviewing and editing, Visualization C.M.P.: Conceptualization, Formal Analysis, Writing–original draft, Writing–reviewing and editing, Supervision, Funding Acquisition.

## Competing interests

The authors declare no competing interests.
