## [Peer Review File · Nature Communications]

HRDE-2 drives small RNA specificity for the nuclear Argonaute protein HRDE-1REVIEWER COMMENTS

Reviewer #1 (Remarks to the Author):

C. elegans is unique in having a greatly expanded Argonaute family. Each Argonaute associates with a specific class of sRNAs, but how this sorting is achieved is not known. This study reports on genetic experiments that identify the novel protein HRDE-2 as required for the sRNA specificity of the nuclear Argonaute HRDE-1.

Major findings include

1. HRDE-1 which is normally nuclear enriches in perinuclear granules when mutated its in sRNA binding site or when in strain defective for the production of WAGO 22G-RNAs, the sRNAs bound by HRDE-1.
2. Localization of HRDE-1 to perinuclear granules in absence of its preferred sRNAs requires HRDE-2, which also enriches in nuclear granules.
3. HRDE-1 and HRDE-2 co-immunoprecipitate from worm extracts and this interaction appears enhanced when HRDE-1 is unloaded.
4. In *hrde-2* mutants, HRDE-1 now interacts with sRNAs that are normally found associated with a different Argonaute CSR-1 – this change in specificity correlates with deposition of the HRDE-1 silencing chromatin mark on CSR-1 targets, but does not appear to have much effect on mRNA levels.

These point are well documented and presented – A table that summarizes these key findings would be welcome as the manuscript is quite dense.

A major assumption throughout this work and in the model presented in Fig. 8 is that recruitment of unloaded HRDE-1 to germ granules by HRDE-2 is what drives specific small RNA loading into HRDE-1. The assumption is that CSR-1 sRNA loading occurs in the cytoplasm and thus, by sequestering unloaded HRDE-1 in granules, HRDE-2 causes HRDE-1 to favor small RNAs generated in germ granules (namely mutator foci). While an attractive model, the authors do not provide direct evidence that localization is necessary for specificity. HRDE-2, like most germ granule proteins, is not exclusively present in germ granules, and is also present diffusively in the cytoplasm. The authors also do not quantify the proportion of HRDE-1 in the different compartments. For the model to work the majority of unloaded HRDE-1 should be in germ granules, which is clearly not the case as shown in Fig. 3. I suggest the authors present the “granule model” as speculative and consider alternatives, such as the formation of specific HRDE-1:HRDE-2 complexes in the cytoplasm.

The authors also do not comment on the organismal consequences on fertility of misloading of HRDE-1? Is HRDE-1 function dispensable for normal germline function? The authors mention a “Mortal germline” phenotype for *hrde-2* mutants but do not explain the significance of this, which is not obvious to non-experts.

Finally, the SIMR-1 data do not add much to the paper and the authors may consider restructuring the paper to focus on HRDE-1 and HRDE-2. How SIMR-1 contributes to HRDE-1 loading is unclear. Unloaded HRDE-1 and HRDE-2 still localize to perinuclear granules even in the absence of SIMR-1. The only evidence that the authors provide to support a function for SIMR-1 in this process is a HRDE-1 IP in the *simr-1* mutant. The data is harder to interpret as both WAGO and CSR small RNAs seem to be enriched in the HRDE-1 IP. Although, the phenotypes of the *hrde-2* and *simr-1* null mutants are similar, the authors need additional data to support this claim.

Reviewer #2 (Remarks to the Author):

In their paper, Chen & Phillips aimed to discover additional components of SIMR-1 foci, a sub-

compartment of cytoplasmic germ granules that their lab had previously identified. They performed a Y2H screen and found that HRDE-2 (heritable RNAi deficient) was a binding partner, which was surprising because the two proteins didn't initially co-immunoprecipitate (co-IP), and SIMR-1 immunoprecipitation followed by mass spectrometry (SIMR-1 IP-MS) did not detect HRDE-2. To support their findings from the Y2H screen, the authors demonstrated that these two proteins do, in fact, colocalize in SIMR foci. To resolve the initial inconsistency, they conducted HRDE-2 IP-MS, which revealed an interaction with nuclear HRDE-1. Further investigation showed that HRDE-1 is present in both the nucleus and germ granules, but its nuclear localization depends on WAGO-class 22G levels. In a *mut-2* background, where WAGO 22Gs are depleted, HRDE-1 relocates from the nucleus to germ granules. Additionally, in the absence of HRDE-2, HRDE-1 no longer concentrates in germ granules, indicating a dependence on HRDE-2 for germ granule localization.

The authors also created a mutant HRDE-1 that is defective in binding to small RNAs (validated biochemically). HRDE-1's biochemical interaction with HRDE-2 was strengthened within SIMR foci in this unbound state. A significant aspect of this study is the analysis of small RNAs bound to HRDE-1 in different genetic backgrounds (wild type, *mut-1*, *hrde-2*, and *mut-2;hrde-2* mutants), revealing that SIMR-1 and HRDE-2 prevent the loading of CSR-class 22G-RNAs into HRDE-1. This finding is noteworthy as it addresses the longstanding challenge of distinguishing between WAGO- and CSR-class 22G RNAs. Since HRDE-1 is known to regulate the H3K9me3-mediated silencing of WAGO-targeted transcripts, the authors investigated whether redirecting CSR-class 22Gs to HRDE-1 would result in the transcriptional silencing of these targets. While H3K9 levels increased in CSR-targets in *hrde-2* mutants, this did not reduce mRNA expression of CSR targets, at least under permissive temperatures, which is a mystery yet to be resolved.

This study presents groundbreaking findings that shed light on the distinction between WAGO- and CSR-class 22G RNAs and their role in transgenerational memory of germline-licensed expression. However, there is one significant issue below.

The colocalization images in Figures 1D and 3E raise concerns, as they indicate a potential chromatic shift or channel movement, impacting granule distance measurements, as all mutator foci appear lower and to the left of HRDE-2 foci.

Reviewer #3 (Remarks to the Author):

Summary:

In this manuscript by Chen and Phillips, the authors initially set out to identify new SIMR foci components by screening for direct protein interactors of SIMR-1 using Yeast Two-Hybrid. In this screen they identified HRDE-2, a protein that had previously been found to localize to germ granules and that was shown to be involved in RNAi inheritance. Once identified, the authors began to study HRDE-2 in more depth and this led to unexpectedly uncovering a role for HRDE-2 in the proper sorting and loading of 22G-RNAs into the Argonaute HRDE-1. The question of how specific small RNAs are loaded into particular AGOs (out of 19 AGOs in total!) in the worm is a longstanding question, and the authors do a careful job of picking apart some of the mechanisms surrounding this. Through a series of complementary genetic, molecular/biochemical, cell biology and genetic approaches, they demonstrate that HRDE-1 localizes to germ granules in a small RNA unbound state, dependent on HRDE-2. Without HRDE-2, HRDE-1 mis-loads CSR-1 (another AGO) class small RNAs, and because it is a nuclear AGO, it is guided by these CSR-1 22G-RNAs to genomic target loci, where it influences histone modifications. This manuscript provides a significant advance in the field's understanding of small RNA sorting into specific AGOs, and shines new mechanistic light onto the functions and dynamics of HRDE-1. However, there are a few concerns and constructive suggestions to address that would improve the manuscript and solidify some of the conclusions.

Major Concerns:

1. It is curious that the interaction observed by Y2H between SIMR-1 and HRDE-2 cannot be recapitulated with orthogonal approaches *in vivo*. I appreciate that the interaction may be transient/highly dynamic or occur with only a small pool of each protein. However, I wondered if the authors attempted any crosslinking-IP experiments to try to capture the interaction *in vivo*. This might be a useful approach to verify the interaction, along with trying to use AlphaFold to model interactions between the two proteins. Nevertheless, although this was the starting point for the project, the fact that the two proteins do not interact *in vivo* does not have much consequence on the major findings of the paper, in our opinion.

2. The authors report that the antibody used for Cut&Tag studies, abcam-ab889b, recognizes H3K9me3 in addition to H3K27me3 and H4K20me3. They were clear and transparent about this, and explicitly stated this caveat (that their analysis may not be limited to H3K9me3), which I appreciated and which is useful for the field to know in general. This "antibody" is a rabbit polyclonal, meaning that there are a mix of antibodies, and that the same peptide is used as an antigen in repeated rounds of antibody production from different rabbits—i.e., what you get as an antibody from one rabbit may not be equivalent to the next rabbit. This is a problem that has long plagued the chromatin field. So, I was left wondering how we can distinguish between differences from one lot/rabbit to the next in antibody specificity versus an effect whereby the antigen used always leads to the production of antibodies that are less specific. I was concerned about this specificity issue and the strength of the conclusions that we can draw using this antibody. While all of the marks are associated with heterochromatin, there are obviously subtle and important functional implications for these different marks, and by far, H3K9me3 has been the biggest focus I think it would be useful to repeat the experiment or instead try ChIP-qPCR with another antibody or several antibodies (perhaps a monoclonal), even with the caveat that the other antibody may also have specificity issues.

3. One major question in our minds related to HRDE-1 loading CSR-1 small RNAs is: what about CSR-1? What small RNAs does it bind under mutant conditions, when HRDE-1 binds its small RNAs? Is it binding to the same small RNAs/targeting the same genes at a lower level (fewer small RNAs), or is it doing something different or unexpected? These questions could be answered with CSR-1 IP/small RNA-sequencing.

4. It occurred to us that in Supplementary Figure 1, there is no way to know which sub-granules HRDE-2 and SIMR-1 localize to in the absence of the other without the presence of other markers like ZNFX-1 or PGL-1, etc., as used elsewhere in the paper. It seems like the strongest statement one could make is that these proteins are still present in granules, but it is not entirely clear which granule these are. On lines 171-173, the authors state: "These data indicate that SIMR-1 and HRDE-2 localize to SIMR foci independently of one another and neither is acting as a scaffolding protein for SIMR foci assembly." In the absence of additional markers, this statement should probably be toned down to "germ granules" instead of SIMR foci.

5. Do the metagene profiles of CSR-1 small RNAs bound to HRDE-1 look the same or different than the pattern of CSR-1 small RNAs? CSR-1 and HRDE-1 small RNA distributions across their target protein coding genes seem somewhat different, so it would be potentially useful to know which pattern prevails in this aberrant situation (*hrde-2* mutant). These patterns might also provide clues about how the small RNAs are made/by which RdRP, since CSR-1 small RNAs are made by EGO-1 in the cytoplasm and HRDE-1 small RNAs are thought to be made by RRF-1 in the Mutator foci. It may be useful to consider using *rrf-1* or *ego-1* mutants (or degron strains) in future studies to help disentangle this relationship, as well.

6. The authors use a *mut-2* mutant to examine localization of "unloaded" HRDE-1 by comparison with other germ granule markers. However, they later report "Even in a *mut-2* mutant, where WAGO-class small RNAs are depleted by 86.2%, HRDE-1 is strongly associated with the remaining small RNAs mapping to WAGO-target genes". Suggesting that even in the *mut-2* background, HRDE-1 remains

loaded with some small RNAs (the localization pattern in Fig. 3, also indicates that small RNAs are loaded, because HRDE-1 is partially nuclear in a *mut-2* mutant). The authors later examine the localization of HRDE-1(HK-AA), the RNA binding deficient mutant, but do not compare its localization to known germ granule markers. Thus the statement "Unloaded HRDE-1 associated with SIMR foci" (line 261) should be revised to say "germ granules" or the authors could directly examine the localization of HRDE-1(HK-AA) in comparison to known germ granule markers.

7. In the text, the authors report proportions of targets that are the targets of HRDE-1 bound small RNAs in wild type and mutant samples (Lines 336-348). It would be useful to know the number of targets and totals, not just proportions here. The authors should consider adding this information to the text to improve clarity and transparency.

8. Related to this point (in 7), "a subset of WAGO-target and spermatogenic genes are normally targeted by HRDE-1 in adult animals, and when that targeting is lost in a *hrde-2* mutant, they show an increase in mRNA expression." Could the authors provide more detail? Which genes are these, how many targets are in this group?

9. Lines 533-535: "However, since we have not directly tested whether the HRDE-1 NLS mutant or IDR deletion mutant binds 22G-RNAs, we cannot definitively determine whether they are siRNA bound." It would be useful to know, by IP-sRNA-seq (or perhaps qPCR) of the NLS and IDR mutants, whether they bind to small RNAs, and if so, which small RNAs they associate with. The prediction is that they do not bind small RNAs, given their localization. It would also be useful to demonstrate that the interaction with HRDE-2 is reduced in these mutants.

Minor changes to text/grammar:

Line 249: "First, to establish whether HRDE-1 forms germ granules when unloaded..."

Considering changing to "First, to establish whether HRDE-1 is recruited to germ granules when unloaded..."

Line 374-5: "Our data demonstrate that HRDE-1 binds to CSR-class small RNAs and localizes to nucleus in a *hrde-2* mutant" add a "the" before nucleus or pluralize nucleus to nuclei

REVIEWER COMMENTS

Reviewer #1 (Remarks to the Author):

C. elegans is unique in having a greatly expanded Argonaute family. Each Argonaute associates with a specific class of sRNAs, but how this sorting is achieved is not known. This study reports on genetic experiments that identify the novel protein HRDE-2 as required for the sRNA specificity of the nuclear Argonaute HRDE-1.

Major findings include

1. HRDE-1 which is normally nuclear enriches in perinuclear granules when mutated its in sRNA binding site or when in strain defective for the production of WAGO 22G-RNAs, the sRNAs bound by HRDE-1.
2. Localization of HRDE-1 to perinuclear granules in absence of its preferred sRNAs requires HRDE-2, which also enriches in nuclear granules.
3. HRDE-1 and HRDE-2 co-immunoprecipitate from worm extracts and this interaction appears enhanced when HRDE-1 is unloaded.
4. In *hrde-2* mutants, HRDE-1 now interacts with sRNAs that are normally found associated with a different Argonaute CSR-1 – this change in specificity correlates with deposition of the HRDE-1 silencing chromatin mark on CSR-1 targets, but does not appear to have much effect on mRNA levels.

These point are well documented and presented – A table that summarizes these key findings would be welcome as the manuscript is quite dense.

We have updated Figure 8A to additionally include a visual representation of HRDE-1 localization and small RNA loading in *mut-2* and *mut-2; hrde-2* mutants. Additionally, we added Supplementary Figure 8 which includes a summary table presenting the key phenotypes (HRDE-1 localization, loading, and bound small RNA class) for wild-type and mutants examined in this manuscript, and a model depicting HRDE-1 loading in a *hrde-2* mutant.

A major assumption throughout this work and in the model presented in Fig. 8 is that recruitment of unloaded HRDE-1 to germ granules by HRDE-2 is what drives specific small RNA loading into HRDE-1. The assumption is that CSR-1 sRNA loading occurs in the cytoplasm and thus, by sequestering unloaded HRDE-1 in granules, HRDE-2 causes HRDE-1 to favor small RNAs generated in germ granules (namely mutator foci). While an attractive model, the authors do not provide direct evidence that localization is necessary for specificity. HRDE-2, like most germ granule proteins, is not exclusively present in germ granules, and is also present diffusively in the cytoplasm. The authors also do not quantify the proportion of HRDE-1 in the different compartments. For the model to work the majority of unloaded HRDE-1 should be in germ granules, which is clearly not the case as shown in Fig. 3. I suggest the authors present the “granule model” as speculative and consider alternatives, such as the formation of specific HRDE-1:HRDE-2 complexes in the cytoplasm.

We emphasize in the discussion that the model is a hypothesis. Additionally, we now provide the alternate possibility that HRDE-2 could promote correct HRDE-1 loading in both the germ granules and cytoplasm, and that further investigation will be necessary to shed light on this possibility (see lines 545-549 of manuscript – tracked changes version).

The authors also do not comment on the organismal consequences on fertility of misloading of HRDE-1? Is HRDE-1 function dispensable for normal germline function? The authors mention a “Mortal germline” phenotype for *hrde-2* mutants but do not explain the significance of this, which is not obvious to non-experts.

We have added a description of the *hrde-1* mutant mortal germline phenotype to the introduction, lines 68-70.

Finally, the SIMR-1 data do not add much to the paper and the authors may consider restructuring the paper to focus on HRDE-1 and HRDE-2. How SIMR-1 contributes to HRDE-1 loading is unclear. Unloaded HRDE-1 and HRDE-2 still localize to perinuclear granules even in the absence of SIMR-1. The only evidence that the authors provide to support a function for SIMR-1 in this process is a HRDE-1 IP in the *simr-1* mutant. The data is harder to interpret as both WAGO and CSR small RNAs seem to be enriched in the HRDE-1 IP. Although, the phenotypes of the *hrde-2* and *simr-1* null mutants are similar, the authors need additional data to support this claim.

We agree with the reviewer that role for HRDE-2 in promoting correct HRDE-1 small RNA loading is more clear than that for SIMR-1. Nonetheless, this project originally came about through our initial effort to identify SIMR foci components, and HRDE-2 and unloaded HRDE-1 clearly localize to SIMR foci. Further, *simr-1* mutants substantially disrupt HRDE-1 small RNA specificity, just not to the extent of *hrde-2*. Thus, we feel that this data is important to include in the manuscript.

In another story that we are working on now, we are additionally examining the role of SIMR-1 in a parallel pathway with NRDE-3 and ENRI-2 in the somatic/embryonic nuclear RNAi pathway. We have preliminary data that suggests SIMR-1 functions similarly in that pathway. Once that story is complete, together these two manuscripts will demonstrate that

SIMR foci are sites of small RNA loading and play key roles in promoting correct small RNA binding for both nuclear Argonaute proteins.

Reviewer #2 (Remarks to the Author):

In their paper, Chen & Phillips aimed to discover additional components of SIMR-1 foci, a sub-compartment of cytoplasmic germ granules that their lab had previously identified. They performed a Y2H screen and found that HRDE-2 (heritable RNAi deficient) was a binding partner, which was surprising because the two proteins didn't initially co-immunoprecipitate (co-IP), and SIMR-1 immunoprecipitation followed by mass spectrometry (SIMR-1 IP-MS) did not detect HRDE-2. To support their findings from the Y2H screen, the authors demonstrated that these two proteins do, in fact, colocalize in SIMR foci. To resolve the initial inconsistency, they conducted HRDE-2 IP-MS, which revealed an interaction with nuclear HRDE-1. Further investigation showed that HRDE-1 is present in both the nucleus and germ granules, but its nuclear localization depends on WAGO-class 22G levels. In a *mut-2* background, where WAGO 22Gs are depleted, HRDE-1 relocates from the nucleus to germ granules. Additionally, in the absence of HRDE-2, HRDE-1 no longer concentrates in germ granules, indicating a dependence on HRDE-2 for germ granule localization.

The authors also created a mutant HRDE-1 that is defective in binding to small RNAs (validated biochemically). HRDE-1's biochemical interaction with HRDE-2 was strengthened within SIMR foci in this unbound state. A significant aspect of this study is the analysis of small RNAs bound to HRDE-1 in different genetic backgrounds (wild type, *mut-1*, *hrde-2*, and *mut-2;hrde-2* mutants), revealing that SIMR-1 and HRDE-2 prevent the loading of CSR-class 22G-RNAs into HRDE-1. This finding is noteworthy as it addresses the longstanding challenge of distinguishing between WAGO- and CSR-class 22G RNAs. Since HRDE-1 is known to regulate the H3K9me3-mediated silencing of WAGO-targeted transcripts, the authors investigated whether redirecting CSR-class 22Gs to HRDE-1 would result in the transcriptional silencing of these targets. While H3K9 levels increased in CSR-targets in *hrde-2* mutants, this did not reduce mRNA expression of CSR targets, at least under permissive temperatures, which is a mystery yet to be resolved.

This study presents groundbreaking findings that shed light on the distinction between WAGO- and CSR-class 22G RNAs and their role in transgenerational memory of germline-licensed expression. However, there is one significant issue below.

The colocalization images in Figures 1D and 3E raise concerns, as they indicate a potential chromatic shift or channel movement, impacting granule distance measurements, as all *mutator* foci appear lower and to the left of HRDE-2 foci.

Reviewer #2's is correct, in noticing that our Deltavision microscope has a chromatic shift between the red and green channels. We corrected for this in the granule distance analysis (Figures 1E and 3F) as detailed in the Materials and Methods – "To account for chromatic shift between channels, distances were calculated between each pair of channels using TransFluorospheres streptavidin-labeled microspheres, 0.04 μm (ThermoFisher, T10711) and these distances were used to correct granule distances." The images shown in Figures 1D and 3E were not corrected, leading to reviewer #2's observation.

To confirm visually that HRDE-2 (or HRDE-1) and SIMR-1 colocalize while *MUT-16* does not, we have repeated the immunostaining in Figures 1D and 3E using a newer Leica Stellaris 5 confocal microscope, utilizing the Lightening deconvolution package. This microscope has higher resolution than the Deltavision and we have replaced the images previously shown in Figures 1D and 3E with this new data. The updated images show that *MUT-16* tends to form smaller granules that are adjacent to SIMR-1, HRDE-2, and HRDE-1, and is consistent with our previous distance quantifications generated on the Deltavision.

Reviewer #3 (Remarks to the Author):

Summary:

In this manuscript by Chen and Phillips, the authors initially set out to identify new SIMR foci components by screening for direct protein interactors of SIMR-1 using Yeast Two-Hybrid. In this screen they identified HRDE-2, a protein that had previously been found to localize to germ granules and that was shown to be involved in RNAi inheritance. Once identified, the authors began to study HRDE-2 in more depth and this led to unexpectedly uncovering a role for HRDE-2 in the proper sorting and loading of 22G-RNAs into the Argonaute HRDE-1. The question of how specific small RNAs are loaded into particular AGOs (out of 19 AGOs in total!) in the worm is a longstanding question, and the authors do a careful job of picking apart some of the mechanisms surrounding this. Through a series of complementary genetic, molecular/biochemical, cell biology and genetic approaches, they demonstrate that HRDE-1 localizes to germ granules in a small RNA unbound state, dependent on HRDE-2. Without HRDE-2, HRDE-1 mis-loads CSR-1 (another AGO) class small RNAs, and because it is a nuclear AGO, it is guided by these CSR-1 22G-RNAs to genomic target loci, where it influences histone modifications. This manuscript provides a significant advance in the field's understanding of small RNA sorting into specific AGOs, and shines new mechanistic light onto the functions and dynamics of HRDE-1. However, there

are a few concerns and constructive suggestions to address that would improve the manuscript and solidify some of the conclusions.

Major Concerns:

1. It is curious that the interaction observed by Y2H between SIMR-1 and HRDE-2 cannot be recapitulated with orthogonal approaches in vivo. I appreciate that the interaction may be transient/highly dynamic or occur with only a small pool of each protein. However, I wondered if the authors attempted any crosslinking-IP experiments to try to capture the interaction in vivo. This might be a useful approach to verify the interaction, along with trying to use AlphaFold to model interactions between the two proteins. Nevertheless, although this was the starting point for the project, the fact that the two proteins do not interact in vivo does not have much consequence on the major findings of the paper, in our opinion.

We attempted to validate this interaction using crosslink-CoIP as suggested, following the crosslink protocol from Du et al., 2023, Cell Reports. However, we were unable to detect an interaction between these two proteins (see below Western Blot result of SIMR-1 IP). We also modeled the interaction between HRDE-2 and SIMR-1 using AlphaFold (see below structures) and identified potential interaction interfaces between these two proteins. Because we have not validated this result in vivo, we do not feel comfortable adding it to the manuscript.

Since HRDE-2 was shown to localize to SIMR foci, we are planning to optimize the TurboID proximal labeling method in our lab, which could potentially capture HRDE-2 as a SIMR-1 proximal protein and shed more light on additional SIMR foci components. We will also explore tagging SIMR-1 with smaller epitopes, such as HA, FLAG, or Myc, at both the N-terminus and C-terminus to rule out any potential interference of HA::mCherry on SIMR-1 interactors.

SIMR-1
HRDE-2

2. The authors report that the antibody used for Cut&Tag studies, abcam-ab889b, recognizes H3K9me3 in addition to H3K27me3 and H4K20me3. They were clear and transparent about this, and explicitly stated this caveat (that their analysis may not be limited to H3K9me3), which I appreciated and which is useful for the field to know in general. This “antibody” is a rabbit polyclonal, meaning that there are a mix of antibodies, and that the same peptide is used as an antigen in repeated rounds of antibody production from different rabbits—i.e., what you get as an antibody from one rabbit may not be equivalent to the next rabbit. This is a problem that has long plagued the chromatin field. So, I was left wondering how we can distinguish between differences from one lot/rabbit to the next in antibody specificity versus an effect whereby the antigen used always leads to the production of antibodies that are less specific. I was concerned about this specificity issue and the strength of the conclusions that we can draw using this antibody. While all of the marks are associated with heterochromatin, there are obviously subtle and important functional implications for these different marks, and by far, H3K9me3 has been the biggest focus I think it would be useful to repeat the experiment or instead try ChIP-qPCR with another antibody or several antibodies (perhaps a monoclonal), even with the caveat that the other antibody may also have specificity issues.

We agree with the reviewer’s concern about the antibody used in our CUT&Tag studies. We acknowledge that this antibody recognizes not only H3K9me3 but also H3K27me3 and H4K20me3. Our transparency about this limitation is intended to provide clarity to the field.

However, unfortunately, there are no commercially available anti-H3K9me3 antibodies that have been demonstrated to be specific for H3K9me3 for CUT&Tag sequencing. We also consulted many researchers in the epigenetics field, and it is a common concern that many commercially available antibodies are not perfect. Therefore, we believe that our choice of using the Abcam-ab889b antibody, which has been widely utilized in the field, including many studies in *C. elegans* HRDE-1 study, is a reasonable choice, despite the crosstalk with other heterochromatin markers. We plan to test additional H3K9me3 antibodies for future experiments but feel that this undertaking is outside the scope of this particular project.

3. One major question in our minds related to HRDE-1 loading CSR-1 small RNAs is: what about CSR-1? What small RNAs does it bind under mutant conditions, when HRDE-1 binds its small RNAs? Is it binding to the same small RNAs/targeting the same genes at a lower level (fewer small RNAs), or is it doing something different or unexpected? These questions could be answered with CSR-1 IP/small RNA-sequencing.

It is interesting to consider whether CSR-1 small RNA binding could be affected by competition with HRDE-1 in the *hrde-2* mutant. It is known, however that *csr-1* mutants are sterile (Claycomb et al., 2009, Cell) and similarly a strain containing a point mutation abrogating CSR-1 small RNA binding is also completely sterile (unpublished work from our lab). Thus, we expect that if HRDE-1’s binding of small RNAs was significantly reducing the number of CSR-class small RNAs available to CSR-1, we would observe more severe fertility and germline morphology defects in the *hrde-2* mutant. Rather, *hrde-2* mutant is fertile at 20°C and sterile at elevated temperature, similar to other mutants in the mutator and inheritance pathways (Sprackin et al 2017, Genetics). It remains possible that there is a modest effect on CSR-1-bound small RNAs, but not one that is significant enough to affect the phenotype of the animals. It is also worth keeping in mind that WAGO-4 also binds the same class of small RNAs, so CSR-1 is already sharing these small RNAs with another Argonaute protein.

4. It occurred to us that in Supplementary Figure 1, there is no way to know which sub-granules HRDE-2 and SIMR-1 localize to in the absence of the other without the presence of other markers like ZNFX-1 or PGL-1, etc., as used elsewhere in the paper. It seems like the strongest statement one could make is that these proteins are still present in granules, but it is not entirely clear which granule these are. On lines 171-173, the authors state: “These data indicate that SIMR-1 and HRDE-2 localize to SIMR foci independently of one another and neither is acting as a scaffolding protein for SIMR foci assembly.” In the absence of additional markers, this statement should probably be toned down to “germ granules” instead of SIMR foci.

We have made the suggested change in the text by stating ‘germ granules’ instead of ‘SIMR foci’.

5. Do the metagene profiles of CSR-1 small RNAs bound to HRDE-1 look the same or different than the pattern of CSR-1 small RNAs? CSR-1 and HRDE-1 small RNA distributions across their target protein coding genes seem somewhat different, so it would be potentially useful to know which pattern prevails in this aberrant situation (*hrde-2* mutant). These patterns might also provide clues about how the small RNAs are made/by which RdRP, since CSR-1 small RNAs are made by EGO-1 in the cytoplasm and HRDE-1 small RNAs are thought to be made by RRF-1 in the Mutator foci. It may be useful to consider using *rrf-1* or *ego-1* mutants (or degron strains) in future studies to help disentangle this relationship, as well.

We have generated the HRDE-1 small RNA metagene profiles as suggested by reviewer #3 and included them in Figure 5. We find that, in the *hrde-2* mutant, HRDE-1 binds small RNAs enriched towards the 3’ ends of CSR-target genes, similar to CSR-1-bound small RNAs. These data are consistent with EGO-1 production of the CSR-class small RNAs

bound by HRDE-1 in *hrde-2* mutants and not with aberrant production of small RNAs from CSR-targets by the *mutator* complex. We have updated the text from lines 358-366 (in tracked changes version of manuscript) and added the metagene plots to Figure. 5C.

We also appreciate the suggestions on utilizing *rnf-1* or *ego-1* mutants or degron strains to study the class of small RNA HRDE-1 binds. In fact, we are currently utilizing these strains to investigate the small RNAs bound by NRDE-3, which will be published in an independent manuscript.

6. The authors use a *mut-2* mutant to examine localization of “unloaded” HRDE-1 by comparison with other germ granule markers. However, they later report “Even in a *mut-2* mutant, where WAGO-class small RNAs are depleted by 86.2%, HRDE-1 is strongly associated with the remaining small RNAs mapping to WAGO-target genes”. Suggesting that even in the *mut-2* background, HRDE-1 remains loaded with some small RNAs (the localization pattern in Fig. 3, also indicates that small RNAs are loaded, because HRDE-1 is partially nuclear in a *mut-2* mutant). The authors later examine the localization of HRDE-1(HK-AA), the RNA binding deficient mutant, but do not compare its localization to known germ granule markers. Thus the statement “Unloaded HRDE-1 associated with SIMR foci” (line 261) should be revised to say “germ granules” or the authors could directly examine the localization of HRDE-1(HK-AA) in comparison to known germ granule markers.

The reviewer is correct that our data indicates that in the *mut-2* mutant, HRDE-1 is partially unloaded and partially loaded with small RNAs. All evidence points to the loaded HRDE-1 being in the nucleus while the unloaded HRDE-1 is retained in granules. Thus, we believe that we are examining the localization of unloaded HRDE-1 when quantifying the distance between portion of HRDE-1 in granules, with the other known granule markers.

7. In the text, the authors report proportions of targets that are the targets of HRDE-1 bound small RNAs in wild type and mutant samples (Lines 336-348). It would be useful to know the number of targets and totals, not just proportions here. The authors should consider adding this information to the text to improve clarity and transparency.

We have clarified the number of targets in the text from lines 373 to 380.

8. Related to this point (in 7), “a subset of WAGO-target and spermatogenic genes are normally targeted by HRDE-1 in adult animals, and when that targeting is lost in a *hrde-2* mutant, they show an increase in mRNA expression.” Could the authors provide more detail? Which genes are these, how many targets are in this group?

We have added additional discussion of the spermatogenic genes in the text from lines 446 to 451, and included the list of genes in the Supplementary Table 6 – Sheet 6.

9. Lines 533-535: “However, since we have not directly tested whether the HRDE-1 NLS mutant or IDR deletion mutant binds 22G-RNAs, we cannot definitively determine whether they are siRNA bound.” It would be useful to know, by IP-sRNA-seq (or perhaps qPCR) of the NLS and IDR mutants, whether they bind to small RNAs, and if so, which small RNAs they associate with. The prediction is that they do not bind small RNAs, given their localization. It would also be useful to demonstrate that the interaction with HRDE-2 is reduced in these mutants.

We appreciate the reviewer’s interest in the HRDE-1 NLS mutant and IDR mutant. We speculate that both HRDE-1 NLS and IDR mutants are loaded, based on the previous research on NRDE-3 (Guang et al, 2008, Science) (see figures below), Guang et al showed that NRDE-3 NLS mutant fails to be translocated to the nucleus (Fig. 3A) while still binds small RNA (Fig. 3C).

To further address whether HRDE-1 NLS mutants are loaded, we generated a strain containing both the NLS (KRS-AAA) mutation and the small RNA binding mutation (HK-AA). We observed that HRDE-1(KRS-AAA; HK-AA) primarily localizes to germ granules, similar to the HRDE-1(HK-AA) small RNA binding mutant. These data indicate that the HRDE-1 NLS mutant (KRS-AAA) is capable of interacting with HRDE-2 in germ granules when empty, and when loaded disperses to the cytoplasm. We expect that the HRDE-1 IDR deletion would behave similarly and we have no reason to expect that either would bind to different small RNAs than the wild-type HRDE-1. We have edited the text from lines 290 to 300 and included the HRDE-1(KRS-AAA; HK-AA) imaging in supplementary figure 3E.

Minor changes to text/grammar:

Line 249: “First, to establish whether HRDE-1 forms germ granules when unloaded...”

Considering changing to “First, to establish whether HRDE-1 is recruited to germ granules when unloaded...”

We have changed the text as suggested.

Line 374-5: “Our data demonstrate that HRDE-1 binds to CSR-class small RNAs and localizes to nucleus in a hrde-2 mutant” add a “the” before nucleus or pluralize nucleus to nuclei.

We have revised the grammar.

REVIEWERS' COMMENTS

Reviewer #1 (Remarks to the Author):

The authors have addressed my comments. In particular they have toned down in the discussion the model that granule association dictates specificity of Argonaute loading and they now propose alternatives. However the title of this paper still states that "germ granule association" drives small RNA specificity when in fact their study does not demonstrate such cause and effect relationship. In my opinion this title is problematic as it further emphasizes a role for granules as "compartments that drive specificity" when in fact the cause and effect could be reversed: specificity (of RNP assembly in the cytoplasm) drives granule formation. I suggest they focus the title instead on the identification of a new Argonaute specificity factor: HRDE-2.

Reviewer #2 (Remarks to the Author):

The authors have sufficiently addressed my critiques concerning the chromatic shift in images 1D and 3E. This is exceptional work and I look forward to its anticipated impact.

Reviewer #3 (Remarks to the Author):

In this revision of the manuscript by Chen and Phillips, the authors addressed the majority of reviewers' concerns satisfactorily. Note that there are some typos/grammatical errors in the newly added sections that can easily be revised. As a final note though, I wanted to caution on making assumptions about AGO activities. While I think that the authors are probably correct on their thoughts regarding CSR-1 misloading/underloading in a hrde-2 mutant, like everything, you don't know for sure until you do the experiment, and the fact that CSR-1 and WAGO-4 share sRNAs is one more reason that CSR-1 might be "under" loaded when HRDE-1 starts to pick up its small RNAs. Perhaps there is even a compensation in the steady state level of CSR-1 22G-RNAs that leads to more being made and allows CSR-1 to maintain its usual loading levels. At any rate, these could be experiments for a future study.

Reviewer #1 (Remarks to the Author):

The authors have addressed my comments. In particular they have toned down in the discussion the model that granule association dictates specificity of Argonaute loading and they now propose alternatives. However the title of this paper still states that "germ granule association" drives small RNA specificity when in fact their study does not demonstrate such cause and effect relationship. In my opinion this title is problematic as it further emphasizes a role for granules as "compartments that drive specificity" when in fact the cause and effect could be reversed: specificity (of RNP assembly in the cytoplasm) drives granule formation. I suggest they focus the title instead on the identification of a new Argonaute specificity factor: HRDE-2.

We thank Reviewer 1's suggestion on the title. We have modified the title to 'HRDE-2 drives small RNA specificity for the nuclear Argonaute protein HRDE-1' to emphasize the identification of HRDE-2 as a Argonaute specificity factor.

Reviewer #2 (Remarks to the Author):

The authors have sufficiently addressed my critiques concerning the chromatic shift in images 1D and 3E. This is exceptional work and I look forward to its anticipated impact.

We appreciate Reviewer 2's acknowledgment of our work.

Reviewer #3 (Remarks to the Author):

In this revision of the manuscript by Chen and Phillips, the authors addressed the majority of reviewers' concerns satisfactorily. Note that there are some typos/grammatical errors in the newly added sections that can easily be revised. As a final note though, I wanted to caution on making assumptions about AGO activities. While I think that the authors are probably correct on their thoughts regarding CSR-1 misloading/underloading in a *hrde-2* mutant, like everything, you don't know for sure until you do the experiment, and the fact that CSR-1 and WAGO-4 share sRNAs is one more reason that CSR-1 might be "under" loaded when HRDE-1 starts to pick up its small RNAs. Perhaps there is even a compensation in the steady state level of CSR-1 22G-RNAs that leads to more being made and allows CSR-1 to maintain its usual loading levels. At any rate, these could be experiments for a future study.

We appreciate Reviewer 3's insightful suggestions to improve the manuscript. We agree that we do not know whether CSR-1 is misloaded/underloaded in the *hrde-2* mutant, and we have added this possibility to the discussion section in line 620-623 (changes tracked version).